



# Nitrogen attenuation, dilution and recycling at the groundwater – surface water interface of a subtropical estuary inferred from the stable isotope composition of nitrate and water

**Authors:** Sébastien Lamontagne[1], Frédéric Cosme[2], Andrew Minard[2], and Andrew Holloway[3]

**Affiliations:**

[1]CSIRO Land & Water, PB 2, Glen Osmond 5064, Australia

[2]Golder Associates, Richmond, VIC 3121, Australia

[3]Golder Associates, St Leonards, NSW 2065, Australia

*Correspondence to*: S. Lamontagne (sebastien.lamontagne@csiro.au)

**Abstract.** Estuarine environments have a dynamic groundwater – surface water interface driven by terrestrial groundwater discharge, tidal cycles, waves and other processes. This interface also corresponds to an active biogeochemical environment. An assessment of discharging groundwater with elevated (>300 mg N L$^{-1}$) NH$_4^+$ and NO$_3^-$ concentrations at such an interface located in a subtropical estuary indicated that 80 % of the N was attenuated, one of the highest N removal rates (>100 mmol m$^{-2}$ day$^{-1}$) measured for intertidal sediments. The

remaining N was also diluted by a factor of two or more by mixing before being discharged to the estuary. Most of the mixing occurred in a 'hyporheic zone' in the upper 50 cm of the riverbed. However, groundwater entering this zone was already partially mixed (12 – 60 %) with surface water via a tidal circulation cell. Below the hyporheic zone (50 – 125 cm below the riverbed), NO$_3^-$ concentrations declined slightly faster than NH$_4^+$ concentrations and $\delta^{15}N_{NO_3}$ and $\delta^{18}O_{NO_3}$ gradually increased, suggesting a co-occurrence of anammox and

denitrification. In the hyporheic zone, $\delta^{15}N_{NO_3}$ continued to become enriched (consistent with either denitrification or anammox) but $\delta^{18}O_{NO_3}$ became more depleted (indicating some nitrification). The discrepancy between $\delta^{15}N_{NO_3}$ (23 – 35‰) and $\delta^{18}O_{NO_3}$ (1.2 – 8.2‰) in all porewater samples indicated that the original synthetic nitrate pool ($\delta^{15}$N ~ 0‰; $\delta^{18}$O ~ 18 – 20‰) had turned-over during transport in the aquifer before reaching the riverbed. Whilst porewater NO$_3^-$ was more $\delta^{18}$O depleted than its synthetic source, porewater

$\delta^{18}O_{H_2O}$ (–3.2 to –1.8‰) was enriched by 1–4‰ relative to rainfall-derived groundwater mixed with seawater. Isotopic fractionation from H$_2$O uptake during the N cycle and H$_2$O production during synthetic NO$_3^-$ reduction are the probable causes for this $\delta^{18}O_{H_2O}$ enrichment.

**Keywords:** groundwater – surface water interactions, submarine groundwater discharge, nitrate, isotopic fractionation





## 1 Introduction

In permeable sediments, there is active mixing between surface water and groundwater by hyporheic exchange and seawater recirculation (Jones and Mulholland, 2000; Heiss and Michael, 2014) (Fig. 1). Hyporheic exchange is induced by flows and currents over uneven riverbeds creating zones where surface water moves in and

porewater moves out of the sediments (Harvey and Bencala, 1993). In marine environments, tides, wave action and density differences between discharging fresh groundwater and seawater also generate groundwater – surface water mixing (collectively referred to as 'seawater recirculation') (Burnett et al., 2003; Sawyer et al., 2013; Precht and Huettel, 2003; Pool et al., 2015). When concentrations are more elevated in groundwater, hyporheic exchange and seawater recirculation can spread a solute load over time and in general will tend to lower concentrations at

the discharge point (Li et al., 1999; Murgulet and Tick, 2016). However, because hyporheic exchange and seawater recirculation also bring labile organic matter, oxygen and other compounds to the subsurface (Santos et al., 2011; Ahmerkamp et al., 2017), mixing zones are also very active biogeochemical environments where contaminants like $NH_4^+$ and $NO_3^-$ can be attenuated via a range of biogeochemical processes (Ullman et al., 2003; Abe et al., 2009; Ueda et al., 2003). When attenuation also takes place, both the contaminant concentration and

the contaminant load to surface water is reduced by groundwater – surface water exchange.

The mixing and attenuation of $NH_4^+$ and $NO_3^-$ in contaminated groundwater discharging into a subtropical estuary was evaluated by collecting riverbed porewater profiles using drive points (Cranswick et al., 2014). A conservative tracer (chloride) evaluated mixing, $^{222}Rn$ estimated residence times (Hoehn and Cirpka, 2006; Lamontagne and Cook, 2007) and various other parameters (including $NH_4^+$ and $NO_3^-$ concentrations and the

isotopic composition of nitrate) evaluated N cycling in the subsurface. The isotopic composition of water was also initially measured to evaluate mixing owing to the large difference in isotopic composition between rainfall-derived groundwater and seawater (Clark and Fritz, 1997). Instead, owing to the large $NO_3^-$ concentrations in this environment, the isotopic composition of water was used to further evaluate N attenuation processes. The implications for the evaluation of the N cycle in contaminated aquifers are discussed.

### 1.1 Key biogeochemical processes

In a contaminated aquifer environment, some of the key processes likely to control the N cycle will include nitrification (Casciotti et al., 2010):

$$NH_4^+ + 2O_2 \rightarrow NO_3^- + H_2O + 2H^+ \tag{1},$$

denitrification (here shown via organic matter oxidation; (Schiff and Anderson, 1987)):

$$(CH_2O)_{106}(NH_3)_{16}(H_3PO_4) + 94.4\ NO_3^- + 92.4H^+ \rightarrow 106CO_2 + 55.2\ N_2 + HPO_4^{-2} + 177.2\ H_2O \tag{2},$$

and anaerobic ammonium oxidation (anammox; (Brunner et al., 2013)):

$$1.3NO_2^- + NH_4^+ \rightarrow N_2 + 0.3NO_3^- + 2H_2O \tag{3}.$$

Annamox tends to co-occur with other biogeochemical processes producing $NO_2^-$, such as denitrification (Zhou et al., 2016). Other possible reactions include ion exchange with aquifer materials, the assimilation of $NH_4^+$ and

$NO_3^-$ into microbial biomass, dissimilatory $NO_3^-$ reduction to $NH_4^+$, and the mineralisation of organic-N during decomposition (Casciotti, 2016; Appelo and Postma, 1993). All the above biogeochemical reactions are expected to modify the nitrogen ($^{15}N:^{14}N$) and oxygen ($^{18}O:^{16}O$) isotope ratios in the original $NH_4^+$ and $NO_3^-$ pools via kinetic fractionation and isotopic equilibrium effects (isotopic ratios are generally expressed in parts per thousands



(‰) relative to a standard using the del ($\delta$) notation or, for $\delta^{15}$N, $(^{15}$N:$^{14}$N$_{sample}/^{15}$N:$^{14}$N$_{standard} - 1) \cdot 1000)$. For example, a $NO_3^-$ pool undergoing denitrification will become more enriched in its heavier isotopes as the lighter ones are selectively removed. The enrichment factor for $\delta^{15}N_{NO_3}$ during denitrification $(^{15}\varepsilon_{NO_3 \rightarrow N_2})$ has been found to vary from $9 - 20$‰ and the one for $\delta^{18}O_{NO_3}$ $(^{18}\varepsilon_{NO_3 \rightarrow N_2})$ from $4 - 16$‰ (Knoller et al., 2011; Bottcher

et al., 1990; Dahnke and Thamdrup, 2013; Wenk et al., 2014). Anammox also strongly fractionates $^{15}$N in the $NH_4^+$, $NO_2^-$ and $NO_3^-$ pools present via kinetic and isotopic equilibrium effects (Brunner et al., 2013). However, the systematics for oxygen fractionation during anammox are not known (Casciotti, 2016). Nitrification is a special case because the $\delta^{18}$O signature of the $NO_3^-$ produced will be a function of the isotopic signature of the ambient $O_2$ and $H_2O$ (Mayer et al., 2001; Snider et al., 2010; Casciotti et al., 2010). Synthetically produced $NO_3^-$

tends to be $^{18}$O-enriched relative to $NO_3^-$ produced via nitrification because in synthetic $NO_3^-$ all the oxygen is atmospheric in origin $(\delta^{18}O_{O_2} \sim 23$‰) whereas during nitrification two out of three O originates from water, which is generally $^{18}$O-depleted $(\delta^{18}O_{H_2O} < 5$‰) relative to atmospheric $O_2$ (Mengis et al., 2001).

### 1.2 Terminology

The following terms are used to define either different sources of water or exchange processes in the profiles.

*Porewater* is used for any water recovered in the subsurface, regardless of its origin. *Terrestrial groundwater* is used for groundwater originating from rainfall recharge before any significant mixing with estuarine water has occurred. The *hyporheic zone* is defined as the upper part of the riverbed where surface and subsurface water mix because of processes such as currents, wave pumping or any other. *Tidal circulation* is the process by which estuarine water tends to move inland over the freshwater table during the rising tide and discharge back to the

estuary during the falling tide. *Surface water* represents the estuary. When describing the profiles, porewater from below the hyporheic zone is further referred to as *groundwater* while porewater within the hyporheic zone is further referred to as *hyporheic water*.

## 2 Methods

### 2.1 Site description

The site is located in the estuarine section of a large river on the east coast of Australia, where an industrial facility is located. Groundwater $NH_4^+$ and $NO_3^-$ concentrations are elevated (>5000 mg N L$^{-1}$ at some locations). The groundwater contamination is widespread at the site and may have several sources. In other words, there is not a single contamination point and associated groundwater plume downgradient. However, the most impacted area is located on the south-eastern side of the site and the associated discharge point along the estuary is known. This

area has been instrumented with nested piezometers transects in the four hydrostratigraphic units present including units 1 and 2, the two most likely to outcrop in the intertidal zone.

Three drive point profiles were collected in the intertidal zone in the vicinity of the main impacted area (Fig. 2). Profile 2 was located in the alignment of the transect of nested piezometers described above, whilst profiles 1 and 3 were approximately located 100 m south and north from Profile 2, respectively. The intertidal

zone at the site consists of a steep artificial rock embankment abutting a silty sand riverbed interspersed with oyster beds on harder substrates. The riverbed would typically only be exposed for a few hours at low tide.



Sampling occurred on the afternoon of 27 April 2017 and was planned to coincide with the monthly low tide level to maximise the window of time available to access the riverbed. Profile 1 was collected at the end of the ebbing tide, Profile 2 at low tide, and Profile 3 during the beginning of the flood tide. The sampling locations were 2 – 5 m offshore from the rock embankment (to prevent interference from buried rocks) and in approximately 1 – 10

5   cm of surface water. Rubber mats were deployed on the riverbed around the drive points to minimise disturbance during sampling. This was only partially successful due to the soft nature of the sediments.

The profiles were collected using a drive point system designed to collect sediment porewater at up to 1.25 m depth below permeable riverbeds. The drive point consisted of a 1.5 m x 24 mm outer diameter stainless steel tube in which 10 cm drive point heads can be screwed on. The drive point heads had 5 cm screens and were

connected to the surface via a 5 mm ID PVC tube in order to minimise the need for purging between samples. The drive points are inserted at suitable depths by gently hammering over a snuggly-fitting brass shoe.

### 2.2 Sample collection

At each location, porewater was sampled at 25 cm intervals from the riverbed surface down to 1.25 m deep (only to 1.0 m at Profile 1). This sampling interval was defined to maintain a high enough vertical resolution to capture

the hyporheic zone while minimising the risk to entrain water from adjacent intervals during sample collection. At each depth, ~65 mL was first purged using a hand-held peristaltic pump. The purge water was used to collect field measurements, including for electrical conductivity, oxygen concentration, pH, temperature, and redox potential using calibrated probes. A further 60 mL was then collected for major ions, 20 mL for $NH_4^+$, $NO_2^-$ and $NO_3^-$, 20 mL for radon-222, 5 mL for the stable isotopes of water, and 40 mL for the stable isotopes of nitrate.

Overall, ~210 mL of porewater was removed at each depth. Assuming a porosity ~0.3 and that porewater was drawn to the drive point from a sphere around the screen, the radius of influence (~6 cm) should not have overlapped between adjacent sampling depths.

Samples for chloride and nutrients were collected in 250 mL plastic containers, stored at 4°C in the field, and 0.45 µm-filtered within a few hours of collection. Samples for stable isotopes of water were 0.45 µm-filtered

in the field and stored in 2 mL vials. Samples for the stable isotopes of nitrate were stored in 60 mL containers and kept at 4°C in the field and frozen within a few hours from collection. Radon-222 samples were collected following the DC method of Leaney and Herczeg (2006). Briefly, the tip of a 20 mL disposable syringe was inserted into the exit tubing from the peristaltic pump and then gently filled by pumping. An initial 6 mL sample was used to flush the syringe and remove air bubbles, followed by a 14-mL sample. The syringe was then fitted

with a 0.45 µm pore-sized disposable filter and needle. The radon sample was preserved by injecting below a mineral oil-scintillant mixture in a pre-weighed scintillation vial.

A surface water sample was collected at the beginning and at the end of the sampling period. Sample collection was as for the drive points, with the exception that field parameters were measured by suspending the probes in the estuary and radon-222 was collected using the PET method (Leaney and Herczeg, 2006) to account

for lower expected radon activities in surface water.

### 2.3 Analytical methods





Chloride and nitrogen species ($NH_4^+$, $NO_3^-$ and $NO_2^-$) were measured by colorimetry at ALS Environmental in Newcastle. The detection limit for chloride and nitrogen species are 1 and 0.01 mg L$^{-1}$ respectively. Stable isotopes of water were sent for analysis at GNS New Zealand and were measured on an Isoprime mass spectrometer; for $\delta^{18}O$ by water equilibration at 25°C using an Aquaprep device, for $\delta^2H$ by reduction at 1100 °C using a Eurovector

Chrome HD elemental analyser. All results are reported with respect to VSMOW2, normalized to internal standards. The analytical precision for this instrument is 0.2‰ for $\delta^{18}O$ and 2.0‰ for $\delta^2H$. The stable isotopes of nitrate were analysed using the bacterial denitrification method to convert $NO_3^-$ into $N_2O$ prior to measurement by isotope ratio mass spectroscopy at Leeder Analytical (Melbourne). Nitrate nitrogen isotope ratios are reported relative to $N_2$ in air and oxygen isotope ratios relative to VSMOW reference water. Internal nitrate isotopic

standards were calibrated to the following standards: IAEA-NO$_3^-$ (+4.7‰$_{air}$, 25.32‰$_{VSMOW}$), USGS32 (+180 ‰$_{air}$, 25.40‰ $_{VSMOW}$), USGS34 (–1.8‰$_{air}$, –27.78‰ $_{VSMOW}$) and USGS 35 (+2.7%$_{air}$, 56.81‰ $_{VSMOW}$). The precision on the nitrogen and oxygen isotopic measurements is 0.5‰. Radon-222 activity was measured by liquid scintillation at CSIRO, Adelaide, with a detection limit of ~3 and 100 mBq L$^{-1}$, for the PET and DC methods, respectively, with a precision of 3 – 5 %.

**2.4 Interpretation**

Separating the role of mixing between groundwater and surface water in the hyporheic zone from the one of attenuation during nitrogen transport in the subsurface can be challenging. A simple graphical approach developed for surface water discharge to estuaries was used to differentiate the contribution between mixing and attenuation in the hyporheic zone (Ullman et al., 2003). In estuaries, river freshwater and associated nutrients mix with

estuarine waters before discharging to the ocean. Tidal cycles in estuaries result in little net movement of water in or out of the estuary but generate significant mixing. This mixing is a dispersive process i.e. solutes tend to move from high to low concentration areas even when no net exchange of water occurs. Similarly, hyporheic exchange can be viewed as a dispersive process in the riverbed, with no net exchange of water but a transport of solutes from high to low concentration areas (Qian et al., 2008). In applying the concept developed for estuaries

to hyporheic exchange, the discharging groundwater flowing through the hyporheic zone is considered to act as the 'river', the hyporheic zone is considered to act as the 'estuary', and surface water is considered to act as the 'ocean' end-member.

     The dynamics of mixing in estuaries have been described by Officer (1979) and Officer and Lynch (1981):

$$F = Qc - K_x A \frac{dc}{dx} \tag{4},$$

where $F$ is the flux of a reactive solute out of the estuary, $Q$ is river discharge, $c$ the solute concentration in the river, $K_x$ the longitudinal dispersion coefficient, $A$ the cross-sectional area of the estuary and $dc/dx$ the solute concentration gradient along the estuary. The approach assumes no density stratification (i.e. the water column is perfectly mixed). At steady-state the distribution of salinity (s; or any other conservative tracer) along the estuary is:

$$Qs - K_x A \frac{ds}{dx} = 0 \tag{5}.$$

Incorporating the two equations, the variations in the reactive solute concentration can be expressed as a function of the variations in salinity:

$$F = Q \left( c - s \frac{dc}{ds} \right) \tag{6}.$$



The advantage of this approach is graphical because the effects of attenuation and mixing can be evaluated visually (Fig. 3). In the case of a solute that is entirely conservative (Line A in Fig. 3), the mixing line is linear. In this case, there is no addition or removal of the solute during transport through the hyporheic zone and the solute flux out is simply $Qc$. However, when a solute is produced in the hyporheic zone (Line B), its concentration

will fall above and, when it is consumed, it will fall below the mixing line (lines C and D). The intercept of the tangent of these curves from the surface water end-member ($c_o*$) is an estimate of the effective solute concentration leaving the hyporheic zone. In other words, $c_o*$ is the concentration that the estuary would 'receive' if there was no mixing in the hyporheic zone, only attenuation. The solute flux out of the hyporheic zone is $Qc_o*$. In the case where $c_o*$ is negative (Line D), all the groundwater input of the solute is consumed within the hyporheic

zone. The negative flux $Qc_o*$ also means that the hyporheic zone is also a sink for solutes imported from the surface water by mixing.

As in the present investigation the hyporheic zone only covered a part of the considered depth profiles, the application of the Officer (1979) model required an adaptation. Below the hyporheic zone, it can be assumed that there is no mixing while attenuation is possible, whereas in the hyporheic zone both mixing and attenuation can

occur. Thus, the profiles were interpreted in two parts: Below (constant salinity) and within the hyporheic zone (variable salinity). The two key assumptions of the application of the Officer (1979) model to the hyporheic zone context is that it is assumed groundwater flow is largely vertical at the scale of the measurements and that concentrations patterns are near steady-state.

## 3 Results

Incidental measurements of water level in the clear tubing of the drive points showed a hydraulic head ~50 cm above river level at 1 m depth in Profile 2 and ~0.2 m above river level at 1.25 m depth in Profile 3. This was consistent with numerous small seeps at the foot of the embankment, showing generalised groundwater discharge in the intertidal zone at low tide.

### 3.1 Field parameters

The three profiles differed markedly in their field parameters (Table 1). Relative to surface water from the estuary, porewater was less saline, especially deeper in the profiles, and also slightly more acidic. Whilst pH was ~7.8 in surface water, it ranged from 6.8 in Profile 2 to 7.1 in Profile 1. The most variable field parameter in the profiles was oxygen concentration. All profiles had declining $O_2$ with depth but over a different range. Profile 1 was well oxygenated throughout (7.8 – 9.2 mg $L^{-1}$), Profile 2 was suboxic (1.9 – 3.1 mg $L^{-1}$) and Profile 3 varied from

suboxic at 1.25 m (1.0 mg $L^{-1}$) to well oxygenated at 25 cm (9.6 mg $L^{-1}$).

### 3.2 Chloride

Based on chloride, the groundwater at the base of the profiles was largely terrestrial groundwater in origin mixed with some estuarine water. The groundwater $Cl^-$ concentration at nested piezometer P3 (~140 m inland; Fig. 2) varies between 28 mg $L^{-1}$ (Unit 1) and 47 mg $L^{-1}$ (Unit 2; A. Minard, *unpublished data*). By comparison, surface

water was 15,000 mg $L^{-1}$ at the time of sampling and porewater at the base of the profiles was 1880, 3290 and



8750 mg $L^{-1}$ for profiles 2, 1 and 3, respectively (Fig. 4a). Thus, at 100 – 125 cm, profiles 1 and 2 were composed of 12 – 20 % surface water while Profile 3 was 60 % surface water. In general, chloride concentrations remained constant between 75 and 125 cm but increased at 50 cm and especially at 25cm, indicating the hyporheic mixing zone was approximately 50 cm in thickness. However, $Cl^-$ concentration remained ~8,000 mg $L^{-1}$ throughout

Profile 3, suggesting a thinner hyporheic zone there. Trends with depth for $\delta^2 H_{H_2O}$ and $\delta^{18}O_{H_2O}$ were similar to chloride but with some subtle differences between profiles. Like for $Cl^-$, the isotopic composition for water indicated mixing with surface water at 25 and 50 cm (Fig. 4b-c). The $Cl^-$ and $\delta^2 H_{H_2O}$ values for profiles 1 and 2 were similar and had lower concentrations relative to Profile 3 but $\delta^{18}O_{H_2O}$ was enriched by ~2‰ in Profile 2 relative to Profile 1 (see additional evaluation for the isotopic composition of water below).

**3.3 Ammonium and nitrate**

There was a general trend for decreasing $NH_4^+$ and $NO_3^-$ concentrations upward in the profiles, but the extent varied materially between profiles (Fig. 4d-e). The highest concentrations were measured in Profile 2 (300 – 400 mg N $L^{-1}$ at the base) and lowest in Profile 3, especially for $NO_3^-$ (0.01 – 1.5 mg N $L^{-1}$). The decline in nitrogen concentration was most pronounced in Profile 2, with $NH_4^+$ and $NO_3^-$ being 53 and 19 mg N $L^{-1}$, respectively at

25 cm. The $NH_4^+$ to $NO_3^-$ ratio tended to increase at shallower depths in Profile 2 and decrease in Profile 3 (Fig. 4f). This means that $NH_4^+$ was lost more rapidly than $NO_3^-$ in Profile 3 while $NO_3^-$ was lost more rapidly than $NH_4^+$ in Profile 2. Nitrite was below detection limit (<0.01 mg N $L^{-1}$) in porewater samples but slightly above detection limit (0.02 and 0.03 mg N $L^{-1}$) in the surface water samples.

**3.4 Stable isotopes of $NO_3^-$**

There was a consistent pattern in the isotopic composition of $NO_3^-$ as a function of depth (Fig. 4g-h). In general, $\delta^{15}N_{NO_3}$ and $\delta^{18}O_{NO_3}$ increased at first from the base of the profiles but tended to decrease once in the hyporheic zone. The increase in $\delta^{15}N_{NO_3}$ and $\delta^{18}O_{NO_3}$ deeper in the profiles would be consistent with the occurrence of a process like denitrification (which leaves the residual $NO_3^-$ pool enriched in its heavier isotopes). However, the decreased $\delta^{15}N$ and $\delta^{18}O$ values once in the hyporheic zone are more difficult to evaluate. These were in part due

to mixing because the isotopic composition of surface water $NO_3^-$ was less enriched than in the porewater. For example, the $\delta^{15}N_{NO_3}$ in surface water was 10.4‰ whilst it varied between 28.2 and 42.1‰ at 50 cm in the profiles. On the other hand, $\delta^{18}O_{NO_3}$ at P1-25 cm was lower than in either deeper porewater or in surface water. This indicates that some nitrate with a low isotopic composition was produced in the hyporheic zone, most likely via nitrification.

**3.5 Radon-222**

The vertical distribution of radon-222 activity indicated a typical pattern for groundwater discharging through a hyporheic zone (Fig 4i). In general, $^{222}$Rn activities ranged between 1–3 Bq $L^{-1}$ in porewater, larger than in surface water (~0.07 Bq $L^{-1}$), and peaked at mid-depth. The lower concentrations in the hyporheic zone are likely due to mixing and the peak at mid-depth can be attributed either to groundwater 'aging' along the flowpath, greater radon

emanation rates from sediments at the edge of the hyporheic zone, or both.





The radon emanation rate from the sediments is not known, so evaluating the apparent age of porewater is more difficult. A minimum groundwater velocity ($v_{low}$) can be estimated by assuming the largest $^{222}$Rn activity measured in the profiles (~3 Bq L$^{-1}$) is close the equilibrium activity with sediments ($A_o$). Below the hyporheic zone (that is, without the need to correct for mixing), the apparent age of porewater can then be estimated from:

$$A_x = A_o\left(1 - e^{-\lambda t}\right) \tag{7},$$

where $A_x$ is the radon activity at a given depth and $\lambda$ the radioactive decay constant for radon. Assuming $A_o \sim 3.5$ Bq L$^{-1}$, the time elapsed between groundwater travelling from 1.25 m to 0.75 m in profiles 2 and 3 would be 4.6 and 7.2 days, respectively, resulting in a $v_{low}$ of 0.11 and 0.07 m day$^{-1}$, respectively. However, the $v_{low}$ estimates assume that the emanation rate is constant with depth, which may not be correct in the vicinity of hyporheic zones because of the potential for $^{226}$Ra (the parent to $^{222}$Rn) to be retained at redox interfaces (Dixon, 1990). Velocity in Profile 1 was not estimated because most samples were in the hyporheic zone, where the additional effect of mixing on radon activities would need to be considered.

An alternative estimate of velocity can be inferred from the hydraulic gradients ($i$) measured during sampling. Using Darcy's Law, velocity would be equal to $K \cdot i/n$, with $K$ the hydraulic conductivity of the sediments and $n$ its effective porosity. $K$ for a silty sand varies between $10^{-7}$ and $10^{-5}$ m s$^{-1}$ (Freeze and Cherry, 1979) and the vertical hydraulic gradients measured at the sites were ~0.2 – 0.5 (at low tide). Assuming an effective porosity ~0.3 and a hydraulic gradient ~0.2 over the tidal cycle, velocity would be 0.006 – 0.6 m day$^{-1}$, overlapping the range found with the radon method. Thus, groundwater travelled through the 1.25 m profiles in two days or more.

### 3.6 Mixing model

In general, there was a net consumption of NH$_4^+$ and NO$_3^-$ in both the groundwater and hyporheic part of the profiles (Fig. 5). In Profile 1, NH$_4^+$ and NO$_3^-$ concentrations fell below the mixing line between groundwater and surface water, indicating consumption in the hyporheic zone. The estimated effective NH$_4^+$ and NO$_3^-$ concentrations ($c_o$*) were 70 and 45 mg N L$^{-1}$, respectively. Thus, the net fraction of N consumed in the hyporheic zone ($f = c_o$*$- c_i/c_i$) was –0.21 and –0.13 for NH$_4^+$ and NO$_3^-$ respectively. In Profile 2, there was a large net consumption of NH$_4^+$ and NO$_3^-$ in both groundwater and in the hyporheic zone. For example, 59 % of the initial NO$_3^-$ at 1.25 m was apparently consumed once groundwater had reached the edge of the hyporheic zone, and a further 87 % of the remaining NO$_3^-$ was then consumed in the hyporheic zone itself. Overall, the attenuation of N in Profile 2 was notable, with the $c_o$* for NH$_4^+$ and NO$_3^-$ being 130 and 22 mg N L$^{-1}$, respectively, relative to concentrations at the base of the profile of 296 and 409 mg N L$^{-1}$, respectively. Thus, ~80 % of the nitrogen load was consumed in the riverbed before discharging to surface water at Profile 2.

In Profile 3, NH$_4^+$ concentrations varied from ~60 mg N L$^{-1}$ at the base of the profile to 48 mg N L$^{-1}$ closer to surface water, so some NH$_4^+$ consumption was likely. As NO$_3^-$ concentrations remained low (<2 mg N L$^{-1}$) throughout the profile, if NH$_4^+$ was consumed by nitrification then denitrification probably occurred as well. The low NO$_3^-$ concentrations are consistent with the low oxygen concentrations in this profile, which would favour denitrification over nitrification. Because NO$_3^-$ concentrations in porewater were generally lower than in the surface water, surface water NO$_3^-$ imported to the subsurface by hyporheic exchange was probably consumed at Profile 3 (i.e. similar to Line D on Fig. 3).

### 3.7 Mixing models for the stable isotopes of nitrate





The variations in porewater $\delta^{15}N_{NO_3}$ and $\delta^{18}O_{NO_3}$ independent of mixing were also evaluated using the estuarine mixing model (Fig. 6). The general trends are similar between profiles. In the 'groundwater' zone (i.e., the part of the profile below the hyporheic zone) both the $\delta^{15}N$ and the $\delta^{18}O$ of $NO_3^-$ became more enriched (more positive) at shallower depths, consistent with the occurrence of processes such as denitrification and anammox. However, in the hyporheic zone the trends were more complex. In general, isotopic enrichment continued at 50 cm but an isotopic depletion was evident at 25 cm, especially for $\delta^{18}O_{NO_3}$. The only exception to this pattern was $\delta^{15}N_{NO_3}$ in Profile 2, where the gradual enrichment persisted across the profile. Thus, attenuation processes appear to have a stratified distribution in the hyporheic zone, with evidence for greater nitrification relative to anammox/denitrification at 25 cm.

### 3.8 Stable isotopes of water

The patterns in the stable isotopes of water in the profiles were unusual and are explored in more detail here. The isotopic signature of terrestrial groundwater at the site has not been measured but should be somewhere between average annual rainfall ($\delta^2H$ = –20.2‰ and $\delta^{18}O$ = –4.50‰) and average winter rainfall ($\delta^2H$ = –33.0‰ and $\delta^{18}O$ = –6.24‰) for Sydney (the closest data available for the site; Hughes and Crawford, 2013). This is consistent with the isotopic signature for shallow groundwater in Sydney ($\delta^2H$ = –22.9‰ and $\delta^{18}O$ = –4.77‰; (Hughes and Crawford, 2013)), which is slightly depleted relative to annual Sydney rainfall. The comparison of chloride and $\delta^2H_{H_2O}$ shows that the porewater samples were within expectations for mixing between two water sources (estuarine water and fresh groundwater derived from rainfall), especially if the groundwater end-member was more similar to winter Sydney rainfall (Fig. 7a). However, when looking at chloride and $\delta^{18}O_{H_2O}$ (Fig. 7b), porewater samples from profiles 2 and 3 were at least 1 – 4‰ enriched relative to conservative mixing lines and more similar to annual than winter Sydney rainfall. The discrepancy was noticeable for profile 2 samples, especially when expressed on a $\delta^2H$-$\delta^{18}O$ plot (Fig. 8). Water table evaporation can shift the isotopic composition of groundwater to the right of the meteoric water line (Clark and Fritz, 1997). However, evaporation would enrich both $\delta^2H_{H_2O}$ and $\delta^{18}O_{H_2O}$ whereas (relative to Sydney groundwater), Profile 2 appeared $\delta^{18}O_{H_2O}$ enriched and possibly slightly $\delta^2H_{H_2O}$ depleted. As Profile 2 is aligned with what is thought to be one of the most impacted groundwater flow lines for the site, the apparent shift in the isotopic composition of water may be related to nitrogen cycling during transport in the aquifer.

There is also some evidence for non-conservative mixing in the isotopic composition of water at the scale of the riverbed. In Profile 2, there was a gradual 1‰ depletion in $\delta^{18}O_{H_2O}$ towards the surface once mixing is accounted for (Fig. 9), mirroring the increase in $\delta^{18}O_{NO_3}$ in the same profile. This enrichment was small but still above the precision for $\delta^{18}O_{H_2O}$ measurements (<0.2‰). This apparent enrichment may be an artefact of groundwater flow being in 2D in the intertidal zone, where different flowpaths with slightly different signatures would be sampled with depth, or of temporal variations in the isotopic signature of surface water. However, in both cases variations in $\delta^{18}O_{H_2O}$ and $\delta^2H_{H_2O}$ would be expected, whereas there was no apparent shift in $\delta^2H_{H_2O}$ once mixing was accounted for (Fig. 9). The variations in $\delta^{18}O_{H_2O}$ in Profile 2 may represent an isotopic shift mediated by the significant N consumption in the riverbed at that location.





**4 Discussion**

Many estuaries are at risk of eutrophication because of excessive N loading from industry, agriculture or other sources (Nixon, 1995; Cosme and Hauschild, 2017). However, a mitigating feature found in many catchments is that groundwater – surface water interactions tend to lower the N load by fostering a biogeochemical environment

where N inputs are attenuated by denitrification or other processes (Gomez-Velez et al., 2015; Heiss et al., 2017; Kim et al., 2017). At the site, up to 80 % on the N load in impacted groundwater is removed at the scale of the riverbed and N concentrations are diluted by a factor of two or more in the subsurface by mixing. There are also at least two scales of mixing at this site. At the larger scale, tidal circulation mixes surface and groundwater at the scale of tens of metres (based on chloride trends in the piezometer network; A. Minard, *unpublished data*),

consistent with findings elsewhere (Pool et al., 2015). The degree of mixing by tidal circulation may be variable in space along the beachface, as suggested by the differences in chloride concentrations at the base of the porewater profiles. At the smaller scale, there was also a 50-cm 'hyporheic'-like mixing zone in the riverbed, where tides, currents and waves would induce surface water to move in and out of the sediments. The extent of attenuation at the larger scale of mixing is not known because sampling focussed at the scale of the riverbed. Thus,

the potential for N attenuation during groundwater – surface water mixing at this site is probably larger than the 80 % of the N load estimated at the riverbed scale. Even when using a low estimate of the vertical groundwater velocity (~0.01 m day$^{-1}$), this represents a very high N removal rate (>100 mmol m$^{-2}$ day$^{-1}$) for permeable intertidal sediments (Schutte et al., 2015).

**4.1 Nitrate attenuation and recycling**

The trends in $\delta^{15}N_{NO_3}$ and $\delta^{18}O_{NO_3}$ suggest N is extensively recycled in the aquifer and in the riverbed. The isotopic signature for groundwater NO$_3^-$ in the source area (that is, 400 m from the river) is consistent with a synthetic NH$_4$NO$_3$ source that has been partially nitrified or denitrified ($\delta^{15}N_{NO_3}$ = –7 to +13‰ and $\delta^{18}O_{NO_3}$ = 13 to 35‰; Fig. 10). However, in the porewater profiles, NO$_3^-$ was $\delta^{15}$N enriched (>20‰) and $\delta^{18}$O depleted (1 – 20‰) relative to groundwater NO$_3^-$ near the source. Thus, near the riverbed the NO$_3^-$ is largely 'recycled' in

origin, either from synthetic NH$_4^+$ that has undergone nitrification, synthetic NO$_3^-$ than was assimilated and later remineralised, or from the mineralisation of 'natural' organic N in sediments or the aquifer (Mengis et al., 2001; Snider et al., 2010; Wong et al., 2014). Biogeochemical cycling would tend to favour $^{15}$N gradually being enriched along the flowpath but for $^{18}$O to be reset with a nitrification signature once all the initial NO$_3^-$ source has been consumed. The fractionation processes during nitrification in aquifers are not well understood but it has been

evaluated in soils and in the marine environment, where $\delta^{18}O_{NO_3}$ is a function of the isotopic signature of the ambient dissolved O$_2$ and H$_2$O, fractionation effects during O uptake, and an isotopic equilibrium between H$_2$O and NO$_2^-$ (Casciotti et al., 2010). However, the isotopic equilibrium effect can probably be ignored as a first approximation because NO$_2^-$ was below detection limit in the profiles. Neglecting equilibrium effects and following Casciotti et al. (2010), the $\delta^{18}O_{NO_3}$ for nitrate produced via nitrification will be:

$$\delta^{18}O_{NO_3} = \frac{1}{3}\delta^{18}O_{H_2O} + \frac{1}{3}\left(\delta^{18}O_2 - {}^{18}\varepsilon_{k,O_2} - {}^{18}\varepsilon_{k,H_2O,1}\right) + \frac{1}{3}\left(\delta^{18}O_{H_2O} - {}^{18}\varepsilon_{k,H_2O,2}\right) \qquad (8),$$

where ${}^{18}\varepsilon_{k,O_2} - {}^{18}\varepsilon_{k,H_2O,1}$ is the combined kinetic fractionation factor during nitritation and ${}^{18}\varepsilon_{k,H_2O,2}$ the kinetic fractionation factor during nitratation. Using a porewater $\delta^{18}O_{H_2O}$ ~ –1‰, ${}^{18}\varepsilon_{k,O_2} - {}^{18}\varepsilon_{k,H_2O,1}$ ~ 30‰, ${}^{18}\varepsilon_{k,H_2O,2}$





~ 15‰ (Casciotti et al., 2010), and assuming $\delta^{18}O_2$ ~ 23‰, the $\delta^{18}O_{NO_3}$ of $NO_3^-$ produced via nitrification in the riverbed would be approximately –8‰. Thus, the inference for enhanced nitrification at P1-25 cm based on the low $\delta^{18}O_{NO_3}$ (–2.4‰) is reasonable.

Below the hyporheic zone, there is a tendency for porewater $\delta^{15}N_{NO_3}$ and $\delta^{18}O_{NO_3}$ to become more enriched during transport to the surface. Such a dual enrichment in the isotopic composition of $NO_3^-$ is commonly found in aquifers undergoing denitrification. The fractionation factors previously found for denitrification ($^{15}\varepsilon_{NO_3 \to N_2}$= 9 – 20‰ and $^{18}\varepsilon_{NO_3 \to N_2}$= 4 – 16‰) indicate $\delta^{15}N_{NO_3}$ should increase faster than $\delta^{18}O_{NO_3}$ in the profiles. However, for porewater below the hyporheic zone the reverse pattern occurred, with $\delta^{18}O_{NO_3}$ increasing ~1.5 times faster than $\delta^{15}N_{NO_3}$. As both $NH_4^+$ and $NO_3^-$ concentrations were elevated (>300 mg N L$^{-1}$) and both were apparently consumed during transport in the profiles, anammox probably occurred. Denitrification and anammox can co-occur because a source of $NO_2^-$ (an intermediate product in denitrification) must be present to fuel anammox (Teixeira et al., 2016). In the marine pelagic zone, anammox yields a residual $NH_4^+$ pool that is [15]N-enriched and a $NO_3^-$ product pool that is also [15]N-enriched because $NO_2^-$ is either converted to $NO_3^-$ or $N_2$ (Brunner et al., 2013). The systematics of the oxygen isotopes during anammox are unknown so the impact this process would have on $\delta^{18}O_{NO_3}$ is unclear. However, the faster increase in $\delta^{18}O_{NO_3}$ relative to $\delta^{15}N_{NO_3}$ in the profiles suggests anammox more strongly fractionates [18]O relative to denitrification. Overall, the shifts in the isotopic composition of $NO_3^-$ during transit through the riverbed were consistent with N attenuation via a combination of denitrification, anammox and nitrification, but other processes such as dissimilatory $NO_3^-$ reduction to $NH_4^+$ are also possible.

**4.2 Stable isotopes of water**

The high loading and turnover of synthetic $NO_3^-$ in the aquifer during transit towards the riverbed apparently shifted the isotopic signature of groundwater. To thoroughly evaluate the processes potentially causing this isotopic shift is well-beyond the scope for this study because the systematics for isotopic fractionation are poorly described for groundwater in general (Green et al., 2010) and unknown for oxygen for anammox in particular (Casciotti, 2016; Brunner et al., 2013). However, some preliminary assessments can be made to judge whether the magnitude of the N transformations in the aquifer can realistically shift the isotopic signature for groundwater.

The past $NH_4^+$ and $NO_3^-$ loadings from the source to the terrestrial groundwater and subsequent transit time before discharge to the river are not known. In other words, what the initial N concentration was at the time of recharge for groundwater discharging to the river at the time of sampling is not known. However, N concentrations in excess of 5000 mg N L$^{-1}$ (>0.35 mol L$^{-1}$) have been recently measured in groundwater near the source area (A. Minard, *unpublished data*), indicating initial N concentrations in groundwater currently discharging to the river could also have been high. Assuming an initial concentration of 5000 mg N-$NO_3^-$ L$^{-1}$ (~1 mol O-$NO_3$ L$^{-1}$) with a $\delta^{18}O_{NO_3}$= 20‰, and an initial groundwater $\delta^{18}O_{H_2O}$= –4.8‰ (or $\delta^{18}O_i$) for discharging groundwater at the time of the study, the shift in isotopic signature if all the O-$NO_3^-$ was converted in O-$H_2O$ during transit would be:

$$m_{tot}\delta^{18}O_{tot} = m_i\delta^{18}O_i + m_{O-NO_3}\delta^{18}O_{NO_3} \tag{9},$$

where $m_{tot}$ is the moles of water in the final unit volume, $m_i$ = the molarity of water (~55.6 mol L$^{-1}$), and $m_{O-NO_3}$ the moles of water produced by the consumption of $NO_3^-$ in the initial litre of reactants. Re-arranging and solving for





$\delta^{18}O_{tot}$ yields (for $m_{O-NO_3}$ = 1 mol L$^{-1}$) an isotopic signature of –4.4‰ or a 0.4‰ enrichment relative to the initial groundwater (noting that the initial volume of water would also have had to increase by 1.8% to account for the new H$_2$O produced). This is smaller than the apparent level of $\delta^{18}O_{H_2O}$ enrichment seen in profiles 2 and 3 (1 – 4‰). In part, this can be explained by a potentially larger NO$_3^-$ concentration at the source. For example, for

$m_{O-NO_3}$ = 4 mol L$^{-1}$ (~20 g N-NO$_3^-$ L$^{-1}$), the expected enrichment would be 1.7‰.

Another possibility is that N cycling also promotes a broader isotopic turnover for the water pool. Many biogeochemical processes consume water, produce water or both. For example, the stoichiometry of denitrification by organic matter (Eq. 2) can also be expressed in terms of the gross amounts of water consumed and produced:

$(CH_2O)_{106}(NH_3)_{16}(H_3PO_4) + 94.4\ NO_3^- + 92.4H^+ + 138\ H_2O \rightarrow 106CO_2 + 55.2\ N_2 + HPO_4^{-2} +$

315.2 $H_2O$                                                                                     (10).

In this case, every mole of O-NO$_3^-$ consumed also consumes ~0.5 mole of water as well as producing ~1.1 mole of new H$_2$O. Water consumption during N cycling typically enriches the remaining water pool (Buchwald et al., 2012; Casciotti et al., 2010). For demonstration purposes, Eq. 9 can be expanded by assuming that for each mole

of O-NO$_3^-$ consumed, 0.5 mole of H$_2$O is also consumed ($m_c$), with a $^{18}\varepsilon_{H_2O\rightarrow Product}$ = 20‰:

$m_{tot}\delta^{18}O_{tot} = m_i\delta^{18}O_i + m_{O-NO_3}\delta^{18}O_{NO_3} - m_c(\delta^{18}O_i - {}^{18}\varepsilon_{H_2O\rightarrow Product})$                       (11).

Including isotopic fractionation during water consumption, for 1 O-NO$_3^-$ mol L$^{-1}$ consumed the shift in groundwater $\delta^{18}O_{H_2O}$ doubles to –3.96‰ (or a ~0.8‰ enrichment) relative to the case with no water consumption. As there are several potential kinetic and equilibrium fractionation effects involving water during N cycling

(Brunner et al., 2013; Casciotti et al., 2010), the magnitude of the $\delta^{18}O_{H_2O}$ enrichment associated with N attenuation at the site could be greater. In particular, whilst isotopic exchange equilibrium between NO$_3^-$ and H$_2$O is extremely slow at neutral pHs (Kaneko and Poulson, 2013), it can be significant when a pool of NO$_2^-$ is present (Casciotti et al., 2010). Despite many uncertainties, the apparent shift in $\delta^{18}O_{H_2O}$ in profiles 2 and 3 relative to expectations for rainfall-derived groundwater can be reasonably accounted for by the elevated synthetic NO$_3^-$ load

and its recycling during transport in the aquifer.

If the input of O-NO$_3^-$ to the aquifer was sufficient to shift the $\delta^{18}O_{H_2O}$, the similar input of NH$_4^+$ could also have shifted the $\delta^2H_{H_2O}$ as NH$_4^+$ was nitrified or converted into N$_2$ by anammox during transport in the aquifer. Synthetic NH$_4^+$ sources appear to have a large range $\delta^2H_{NH_4}$ (~60‰; (Benson et al., 2009)), so the potential exists for a large difference in $\delta^2$H content between synthetic NH$_4^+$ and ambient groundwater at the site.

However, the porewater $\delta^2H_{H_2O}$ is within expectations for mixing between winter Sydney rainfall and seawater (but not for Sydney groundwater and seawater). Thus, there is either no effect on $\delta^2H_{H_2O}$ from a large NH$_4^+$ loading or the shift was relatively small at the site. A search of the literature failed to yield any information on $\delta^2H_{NH_4}$ in the environment, so further evaluation of how a high NH$_4^+$ loading could have impacted on porewater $\delta^2H_{H_2O}$ is not possible at present.

**5 Conclusion**

This study demonstrated a strong potential for N attenuation at the groundwater – surface water interface for contaminated groundwater discharging to a subtropical estuary. This finding is consistent with the literature,





where this interface is considered an active environment for dilution of incoming groundwater solutes (Sawyer et al., 2013; Li et al., 1999) and for biogeochemical processes, in particular for nitrogen cycle (Jones and Mulholland 2000; Ullman et al. 2003; Gomez-Velez et al. 2015). However, in an estuarine setting, different scales of groundwater – surface water mixing are present and may synergistically contribute to N attenuation. Much of the

5  N attenuation at the site was probably via anammox, perhaps owing to the unusual composition of the contaminated groundwater (with near molar equivalents of $NH_4^+$ and $NO_3^-$ at the source). Once the systematics of oxygen isotope exchange during the N cycle in aquifers are better understood, the shifts in the isotopic composition of groundwater along a flowpath could become a useful tool to evaluate N attenuation.

### Acknowledgements

10  Sheree Woodroffe and Antony Taylor facilitated access to the study area. Comments by Axel Suckow, Nina Werti and Michael Donn greatly improved earlier versions of the manuscript.

### Disclosure

This study was funded by the company on which property the contaminated groundwater source is located. Site location and the data set can only be provided with the consent of the company.

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





**Table 1.** Field parameters collected in porewater and surface water in the intertidal zone.

| Sample | pH | Electrical conductivity (mS cm$^{-1}$) | Redox potential (mV) | Dissolved oxygen (mg L$^{-1}$) |
|---|---|---|---|---|
| P1-25 cm | 7.05 | 34.8 | 155 | 9.2 |
| P1-50 cm | 7.12 | 13.41 | 129 | 9.16 |
| P1-75 cm | 7.07 | 16.12 | 131 | 7.01 |
| P1-100 cm | 7.2 | 14.21 | 131 | 7.8 |
| | | | | |
| P2-25 cm | 6.84 | – | 155 | 3.09 |
| P2-50 cm | 6.79 | – | 142 | 3.74 |
| P2-75 cm | 6.81 | – | 153 | 2.71 |
| P2 -100 cm | 6.84 | – | 162 | 2.62 |
| P2-125 cm | 6.7 | – | 170 | 1.91 |
| | | | | |
| P3-25 cm | 6.89 | – | 155 | 9.63 |
| P3-50 cm | 7.01 | – | 154 | 3.01 |
| P3-75 cm | 6.96 | – | 154 | 2.82 |
| P3-100 cm | 7.01 | – | 156 | 1.9 |
| P3-125 cm | 7 | – | 157 | 0.95 |
| | | | | |
| Surface water -1 | 7.88 | 74.5 | 195 | 7.74 |
| Surface water -2 | 7.77 | – | 167 | 7.12 |




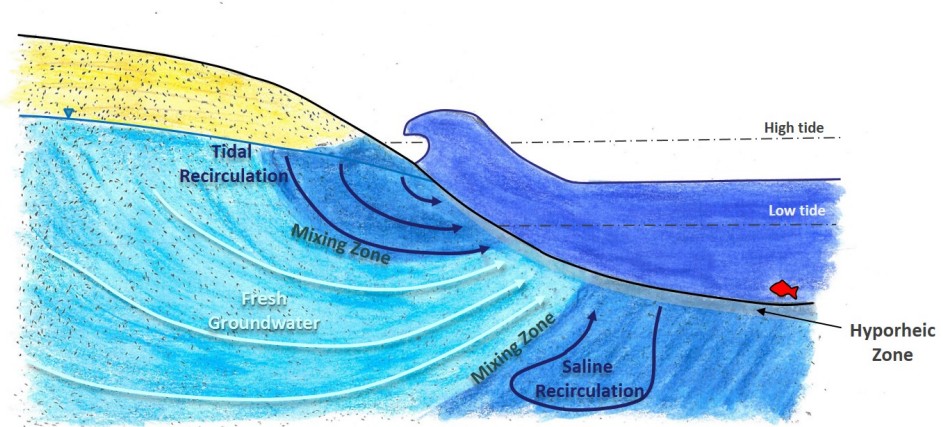

**Figure 1**. Conceptual representation of the different scales of groundwater – surface water mixing in the intertidal zone (modified from (Heiss and Michael, 2014)).



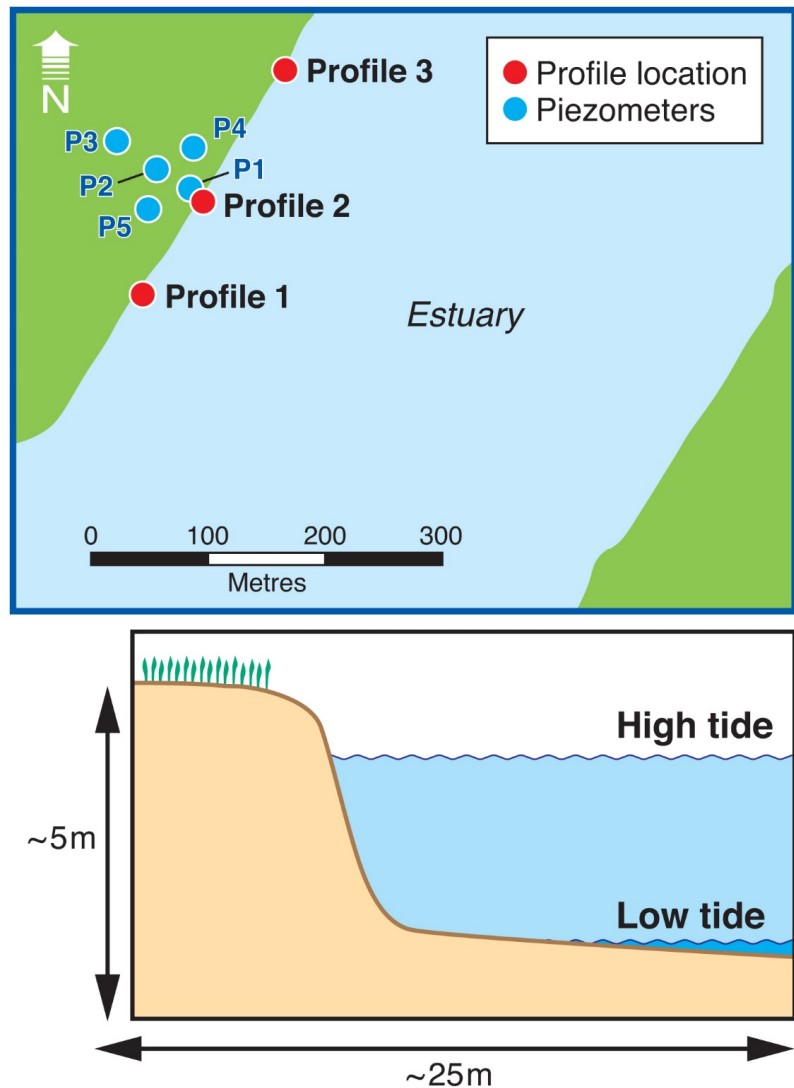

**Figure 2.** Location of the porewater profiles (top) and a schematic cross-section on the intertidal zone (bottom). Also indicated is part of the piezometer network previously installed at the site, approximately aligned with the zone with the most contaminated groundwater.




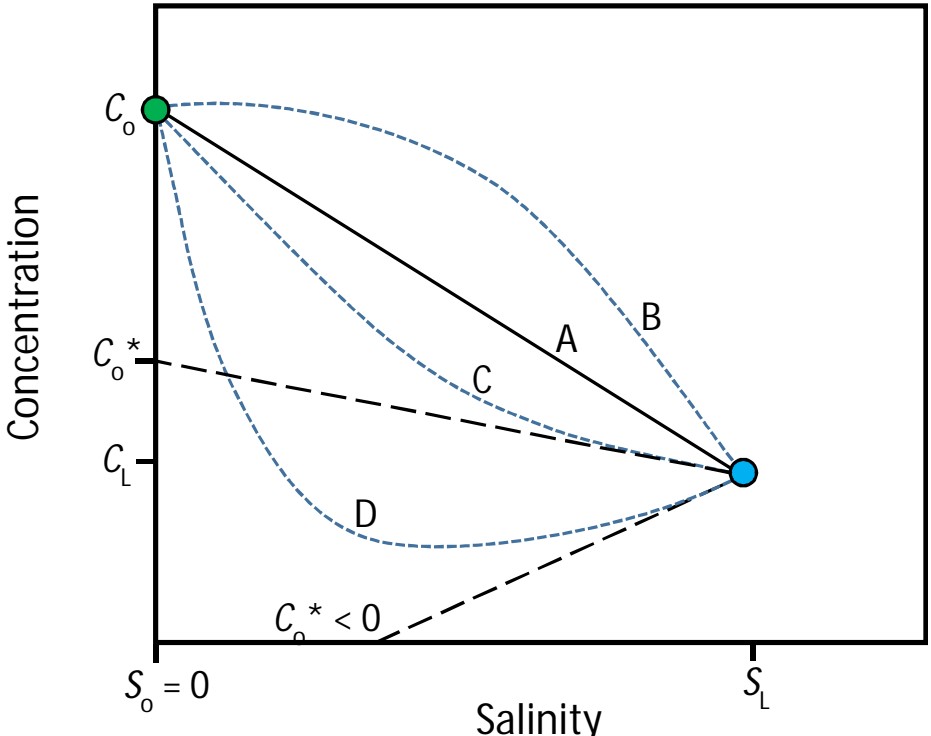

**Figure 3.** Variations in the concentration of a solute along a salinity gradient in an estuary. $c_L$, $s_L$ – solute concentration and salinity in the sea; $c_o$, $s_o$ – solute concentration and salinity in the river; Line A – mixing only (no solute production or consumption in the estuary). Line B – solute production in the estuary. Lines C and D – solute consumption in the estuary; $c_o^*$ – effective solute concentration. For Line D, $c_o^* < 0$ which means the estuary is also a sink for solutes imported by mixing from the sea. Modified from Officer (1979).



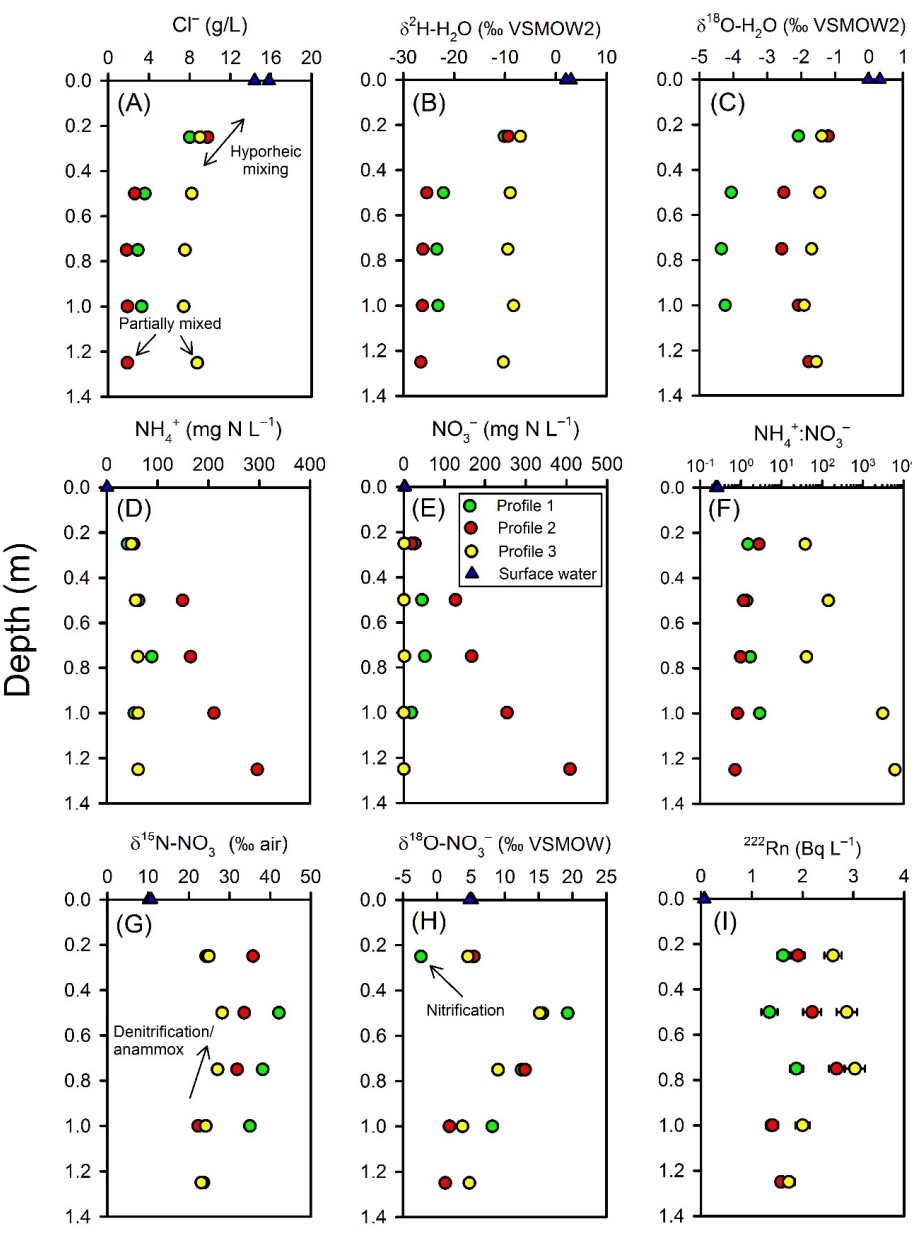

**Figure 4.** Porewater profiles for selected parameters collected in the intertidal zone.

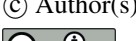



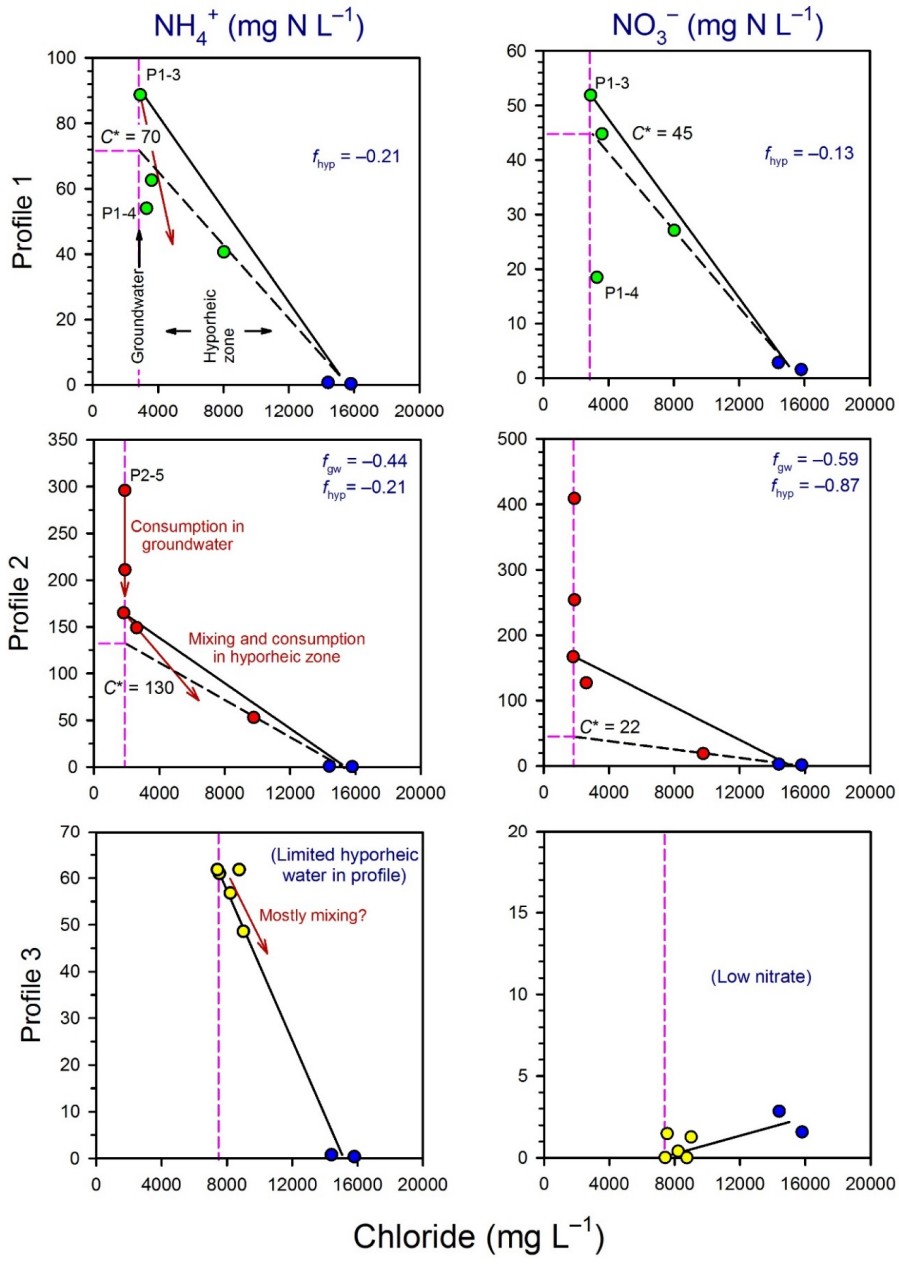

**Figure 5.** Evaluation of mixing and transformations for $NH_4^+$ (y-axes, left column) and $NO_3^-$ (y-axes, right column) for Profile 1 (top), Profile 2 (middle) and Profile 3 (bottom). The vertical pink lines represent samples collected in the 'groundwater' zone and those to the right of this line are within the hyporheic zone, based on chloride concentrations. The blue circles represent the surface water samples.

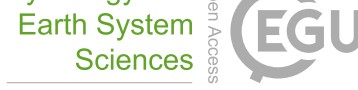

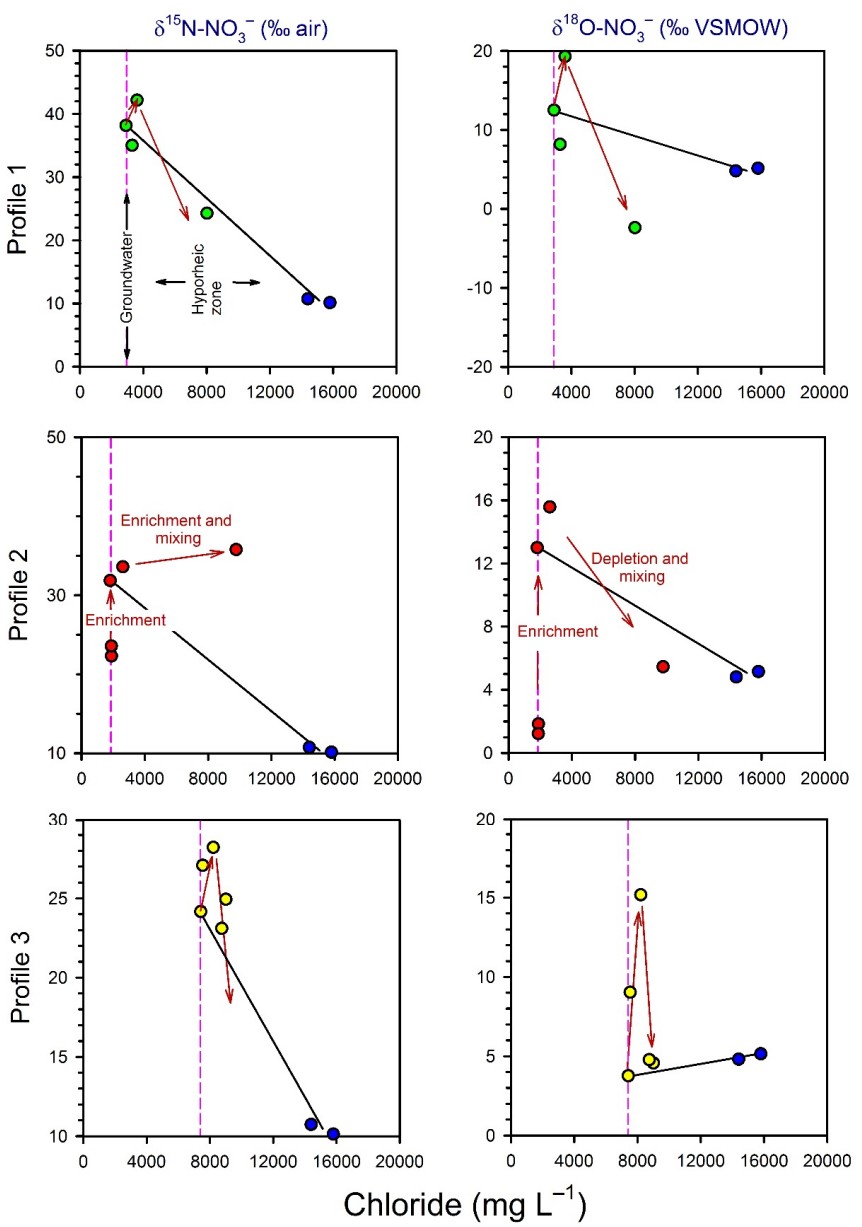

**Figure 6.** Evaluation of mixing and transformations for $\delta^{15}N_{NO_3}$ (y-axes, left column) and $\delta^{18}O_{NO_3}$ (y-axes, right column) for Profile 1 (top), Profile 2 (middle) and Profile 3 (bottom). The vertical pink lines represent samples collected in the 'groundwater' zone and those to the right of this line are within the hyporheic zone, based on chloride concentrations. The blue circles represent the surface water samples.





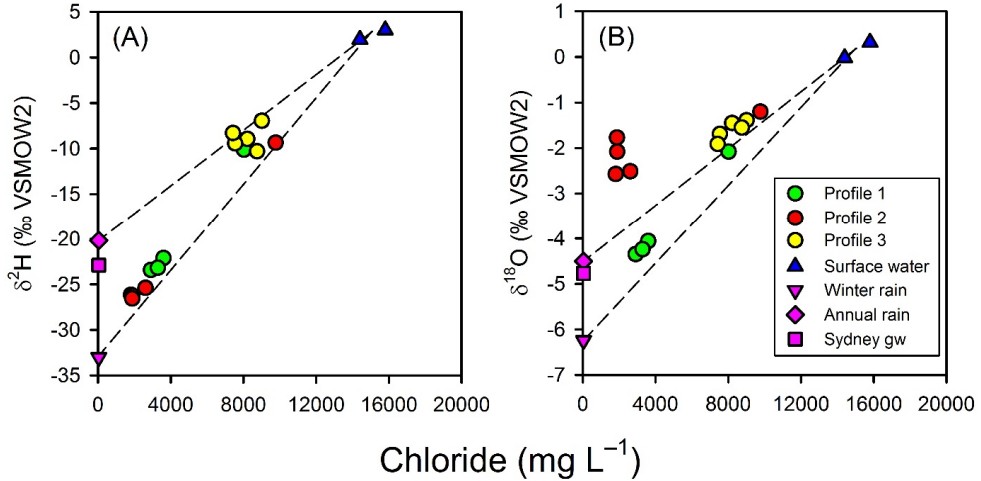

**Figure 7.** Chloride-based mixing lines for the isotopic composition of porewater relative to surface water,
annual Sydney rainfall, and winter Sydney rainfall. Sydney rainfall was used as a proxy for the isotopic
composition of unimpacted groundwater at the site. Note that the isotopic composition of Sydney groundwater

5    is intermediate between annual and winter Sydney rainfall.




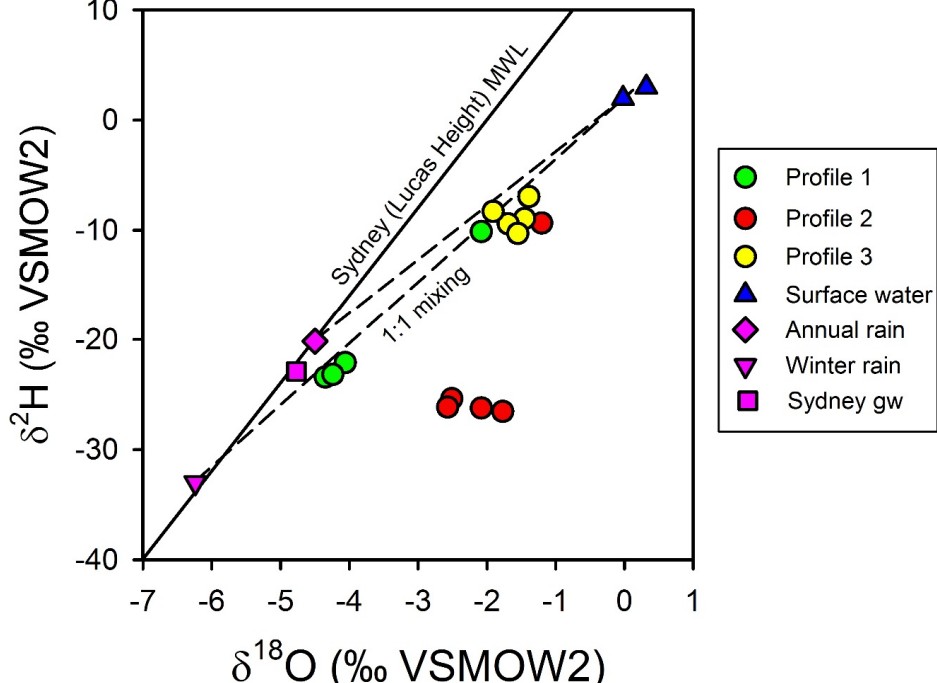

**Figure 8.** Isotopic composition for surface water, porewater, and Sydney groundwater and rainfall relative to the meteoric water line for Sydney (Lucas Height). Dotted lines represent potential mixing lines between fresh groundwater and surface water. Note that evaporation lines for groundwater would be very similar to the mixing line in this environment.




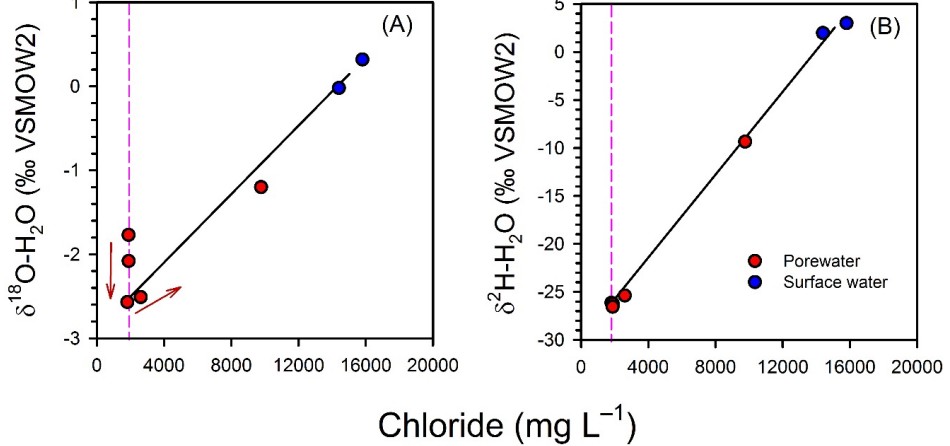

**Figure 9.** Mixing model for the isotopic composition of water for Profile 2, showing a small depletion trend upward in the profiles for $\delta^{18}O_{H_2O}$ (A) but not $\delta^2 H_{H_2O}$ (B).




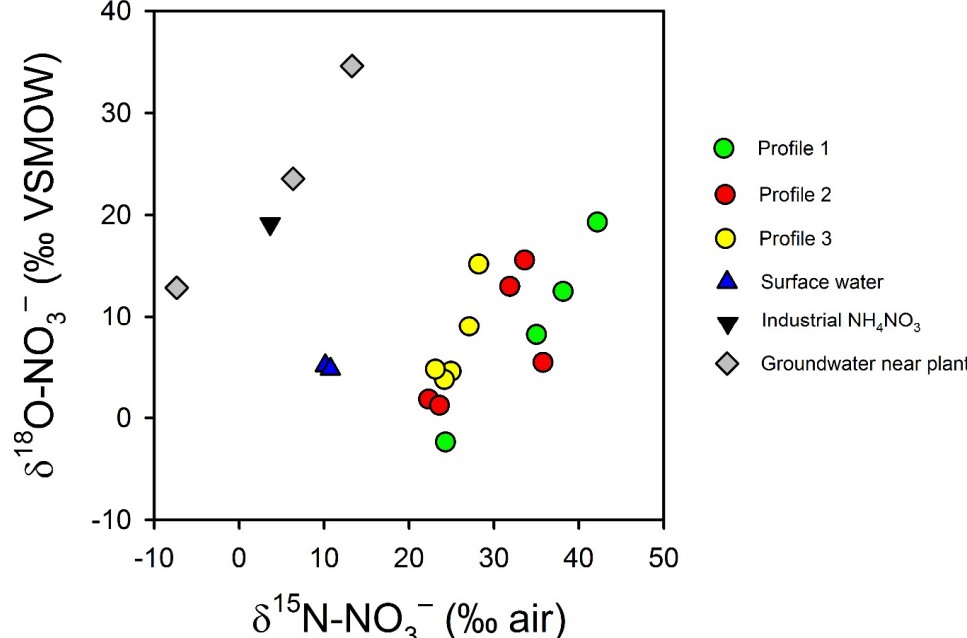

**Figure 10.** Variations in $\delta^{15}N_{NO_3}$ and $\delta^{18}O_{NO_3}$ in intertidal porewater, surface water, one synthetic $NH_4NO_3$ sample from the plant and three groundwater samples collected underneath the plant (A. Minard, *unpublished data*).