# Peer review of "Nitrogen attenuation, dilution and recycling at the groundwater – surface water interface of a subtropical estuary inferred from the stable isotope composition of nitrate and water"

_Hydrology and Earth System Sciences, 2018_

## Referee Comment (RC1) · Anonymous Referee #1 · 16 Apr 2018

Review of Lamontagne et al: Nitrogen attenuation, dilution and recycling at the groundwater – surface water interface of a subtropical estuary inferred from the stable isotope composition of nitrate and water

Lamontagne et al present a manuscript wherein isotopes of oxygen and nitrogen in nitrate in ground water and the ground water – surface water mixing zone are used to estimate nitrogen removal rates downstream of a contamination source. Overall, I find this an interesting application of isotope effects and a potentially useful method. Generally, the methodology looks OK, however there are too many details missing from

the manuscript to judge this certainly. At this stage, the manuscript is not suitable for publication, however I would be very happy to perform a more complete review of this study when a complete manuscript is presented.

Major comments

1. Disclosure of location & data.

I am not comfortable with the lack of disclosure in this MS. For example, it is not appropriate to be unable to give the location of the site. The specifics of the site (size, catchment, land use) are important for placing it in a local, national and global context. In addition, the non-availability of (1) much of the complete dataset and (2) several pieces of data cited as "unpublished" is in conflict with (a) the HESS guidelines on data availability (https://www.hydrology-and-earth-system-sciences.net/about/data_policy.html); (b) modern approaches to data availability in scientific literature (e.g. the general policies of EGU, AGU, ACS); and (c) the FORCE 11 guidelines. I do not accept that this work can serve as (presumably) a private consultation for industry, with confidential outcomes, and also a public, fully peer-reviewed scientific research. The objectives and interests of these two are in conflict. The authors should obtain unconditional permission from the funding organisation to publish the complete dataset and site details.

2. DNRA

DNRA is known to be dominant in many Australian sub-tropical estuaries (e.g. see Dunn refs below and the references therein) – this should not be ignored in your analysis.

DNRA likely produces 15N-depleted NO3, similar to denitrification, but does not result in removal. Thus, it seems somewhat pre-determined to only include removal process (the model assumes denitrification and anammox dominate) and exclude DNRA.

This should be at least addressed.

Dunn, R. J., D. Robertson, P. R. Teasdale, N. J. Waltham, and D. T. Welsh (2013),

[Figure]

Benthic metabolism and nitrogen dynamics in an urbanised tidal creek: Domination of DNRA over denitrification as a nitrate reduction pathway, Estuarine, Coastal and Shelf Science, 131, 271-281.

Dunn, R. J. K., D. T. Welsh, M. A. Jordan, N. J. Waltham, C. J. Lemckert, and P. R. Teasdale (2012), Benthic metabolism and nitrogen dynamics in a sub-tropical coastal lagoon: Microphytobenthos stimulate nitrification and nitrate reduction through photo-synthetic oxygen evolution, Estuarine, Coastal and Shelf Science, 113, 272-282.

Minor comments

3.25 – what River?

4.5 – if the rubber mats were not completely successful, what is the impact and extent of contamination? Discuss.

4.25 – nitrate not filtered for isotopes?

9.14 – it is a pity that you did not measure the rainfall signature or any ground water at the site. It is hard to evaluate how appropriate the Lucas Heights data is, as we do not know where your site is: if it's in south Sydney, this makes sense. If it is in Newcastle or Wollongong, it's tenuous. If it is in Bateman's Bay, this is a useless comparison.

Fig 2 – show direction to sea.

Fig 4 – needs better explanation in caption

Fig 5 – needs to be made simpler. Figure panel lettering would aid in a clearer caption. These figures are busy and unclear. Remove the text rom the figures to discussion.

Fig 6 – see Fig 5.

---

## Referee Comment (RC2) · Anonymous Referee #2 · 18 Apr 2018

**Reviewer's Report**
**EGU Hydrology and Earth System Sciences**
**For the Editors**

| Title | Nitrogen attenuation, dilution, and recycling at the groundwater-surface water interface of a subtropical estuary inferred from the stable isotope composition of nitrate and water |
|---|---|
| Author(s) | Sébastien Lamontagne, Frédéric Cosme, Andrew Minard, and Andrew Holloway |
| Journal | EUG Hydrology and Earth System Sciences |

**EGU Numerical Evaluation by the Reviewer**

| | Excellent (1) | Good (2) | Fair (3) | Poor (4) |
|---|:---:|:---:|:---:|:---:|
| **Scientific significance:** Does the manuscript represent a substantial contribution to scientific progress within the scope of Hydrology and Earth System Sciences (substantial new concepts, ideas, methods, or data)? | X | | | |
| **Scientific quality:** Are the scientific approach and applied methods valid? Are the results discussed in an appropriate and balanced way (consideration of related work, including appropriate references)? | | X | | |
| **Presentation quality:** Are the scientific results and conclusions presented in a clear, concise, and well-structured way (number and quality of figures/tables, appropriate use of English language)? | | X | | |

This is the first paper that I have seen that focuses on a combined hyporheic and intertidal mixing zone and the possible biogeochemical interactions that might occur in this zone. The paper also demonstrates that some of the techniques that are used to study these zones separately might be used to characterize the biogeochemistry of a zone with mixed processes, as the dominant physical process for both of these zones is mixing. I think that the authors could do a better job of emphasizing the novelty of their field setting and the novelty of their approach.

This is also the first environmental science paper that I have seen in over thirty years in the field where the location of the field site is intentionally not provided by the authors. I understand that the funders of this research may have reason to want the location to be obscured, but I think that the authors, perhaps with the assistance of HESS editors, need to convince these patrons to allow the setting to be identified. I am not sure that HESS editors should even permit this article to be published, as novel and as interesting as it is, without the location information. Certainly, the lack of an identified field location dramatically will reduce the scientific utility and impact of the

submission.  Personally, I would be reluctant to cite this paper, given the lack of a well-characterized and identified field site.

Given that the most novel aspect of this paper is how the hyporheic processes in the tidal river interact with the intertidal processes of the estuarine boundary, the tidal should reflect this novelty. I suggest that the title be changed slightly to:  "Nitrogen attenuation, dilution, and recycling in an intertidal hyporheic zone *somewhere, not identified, in* Australia" (Note: the italicized part of the title should be changed to reflect the actual location, if at all possible).

Lastly, I think that the conceptual illustrations provided with this manuscript are excellent and with a few modifications (to indicate both the longitudinal as well as the cross-sectional geometries) these illustrations are likely to stimulate more interest in both hyporheic and intertidal mixing and exchange.

| Title | Nitrogen attenuation, dilution, and recycling at the groundwater-surface water interface of a subtropical estuary inferred from the stable isotope composition of nitrate and water |
|---|---|
| Author(s) | Sébastien Lamontagne, Frédéric Cosme, Andrew Minard, and Andrew Holloway |
| Journal | EUG Hydrology and Earth System Sciences |

This is the first paper that I have seen that focuses on the biogeochemistry of a combined hyporheic and intertidal mixing zone at the margins of an estuarine tidal channel and the biogeochemical interactions that occur in this zone. Hyporheic exchange is the entrainment and exchange fluids between a streambed and a channel due to spatial and temporal changes in pore pressures driven by streamflow and stream topography. Intertidal mixing is the exchange of surface and porewaters in a typically porous (sandy) beach due to tides and waves. Both processes lead to mixing between surface and porewaters. And both processes must occur and interact in estuarine tidal channels; the relative importance of the two processes is probably related to the channel flow rates, stream topography, intertidal topography and tidal range.

Beachfaces and stream margins are also zones of high biogeochemical reactivity due to the availability and interaction of fresh and reactive organic matter from primary productivity, and oxidants, such as $O_2$ and $NO_3^-$. These zones may serve to mitigate the effects of nutrient contamination on potentially fragile coastal ecosystems, in a manner analogous to the use of slow sand filters in wastewater processing.

The paper further demonstrates that some of the techniques that are used to study these zones separately (such as deviations from linearity of reactive property vs conservative property diagrams that are often used to study the biogeochemistry of more traditional estuarine mixing zones.) This is a novel observation and the authors could do a better job of describing and emphasizing the novelty of both their field setting and their approach.

This is also the first environmental science paper that I have seen in over thirty years in the field where the location of the field site is intentionally not provided by the authors. I understand that the funders of this research may have reason to want the location to be obscured, but I think that the authors, perhaps with the assistance of HESS editors, need to convince these patrons to allow the setting to be identified. I am not sure that HESS editors should even permit this article to be published, as novel and as interesting as it is, without the location information or, at least, some agreement that location information will be provided to researchers in the future. Certainly, the lack of an identified field location will dramatically reduce the scientific utility and impact of the submission. Personally, I would be reluctant to cite this paper, given the lack of a well-characterized and identified field site. I am also concerned that the patrons of this work are interfering with the normally "open" scientific process. I urge the authors and editors to seek permission from the patrons of this research to disclose the site location.

Given that the most novel aspect of this paper is how the hyporheic processes in the tidal river interact with the intertidal processes of the estuarine boundary, the tidal should reflect this novelty. I suggest that the title be changed slightly to: "Nitrogen attenuation, dilution, and recycling in an intertidal hyporheic zone *somewhere, not identified,* in Australia" (Note: the italicized part of the title should be changed to reflect the actual location, if at all possible).

I have recommended some additional editorial modifications to the authors. With these modifications, the paper should be acceptable for publication, particularly if the problem of "site-anonymity" can be resolved to the editors' satisfaction. Certainly this work, once and if it is published, will serve as a model for future biogeochemical and biogeophysical studies of similar marginal marine zones.

---

## Author Comment (AC1) · 28 May 2018

We would like to thank the reviewer for his/her comments and we offer to address his/her concerns as outlined below.

1. Disclosure and data location: Not revealing the exact site location struck a nerve with both reviewers. We agree that this is not ideal but have to respect the wishes of the funder and landholder in this regard. The exact site location is frequently not revealed in scientific papers from contaminated sites (see Journal of Contaminant Hydrology,

for example). We have discussed the matter with the HESS Editors and have come up with a compromise. We will provide additional details about the physical setting and add a statement in the Introduction about the requirement from the landholder not to have the exact location revealed. If there is a need to revisit the site by third parties for additional measurements, that could be arranged through a discussion with the landholder. This is one of our motivations to publish this work, to get a second opinion from our colleagues about the potential processes operating at this site. Likewise, we can also arrange to have some of the data referred to as 'unpublished' accessible if required.

2) DNRA: We do list DNRA as one of the other mechanisms that may be operating at the site (line 2.35) but indeed do not evaluate how this process could impact on isotopic signatures in any detail. Because both NH4+ and NO3– are consumed within the profiles, we put the emphasis on denitrification and anammox. However, it is not unreasonable to expect some level of DNRA at the site. We propose to address this by; 1) including DNRA more explicitly in Section 1.1 (review of key biogeochemical processes in intertidal environments), 2) referring to the literature on DNRA more extensively, and 3) discussing how the occurrence of DNRA would impact on isotope signatures.

Minor comments

3.25 What river?: As discussed above we cannot reveal the exact site location, including the river's name.

4.5: Rubber mats: During a preliminary trial, we had observed that the substrate was generally soft underfoot, possibly owing to the large upward hydraulic gradients. This was addressed by minimising movement to a minimum, deploying rubber mats, using harder substrate nearby whenever possible, and using a fairly coarse vertical spatial discretization (25 cm). There was still a noticeable disturbance of the sediments by the end of profiling but it was minimal at the drive point itself. A more detailed look at the scale of the hyporheic zone (cm) would require a different approach but we are

reasonably confident that at the scale of our measurements (tens of cm) the level of disturbance was acceptable.

4.25. Filtration of nitrate samples: The author responsible for the preparation of nitrate samples is away at present but we will clarify this in the revision.

9.24. Isotopic signature for groundwater: It would indeed be preferable to know the isotopic composition of 'pristine' fresh groundwater at the site. The isotopic composition for groundwater further inland at the site has not been measured (yet). However, because N contamination is widespread at the site, determining the 'pristine' isotopic signature could be difficult anyway. The next preferable option would be local rainfall, but we unfortunately only realised that groundwater isotopic composition may not be conservative at the site late in the data interpretation stage. To develop a local meteoric water line for the site would also require a monitoring program of several years. The use of Sydney precipitation and groundwater is the third best option and the only practical one at this stage. The site is reasonably close to Sydney to use it as a proxy. We think we are providing reasonable evidence for groundwater not to be conservative at this site. We are planning to revisit this issue in more detail in the future.

Figures: We will adjust the figures according to the suggestions from both reviewers.

---

## Author Comment (AC2) · 28 May 2018

We thank Reviewer #2 for his comments and we think we can address them in the revision.

1) First study looking at 'hyporheic exchange' and tidal circulation in an estuary: We thank the reviewer for this observation and we will highlight the novelty of this work in the revision.

2) Site location: Please see reply to Reviewer #1 for how we propose to address this

issue.

3) Title: Following the reviewer's suggestion, we propose to modify the title to: "Nitrogen attenuation, dilution and recycling in the intertidal hyporheic zone of a subtropical estuary".

4) We thank the reviewer for additional suggestions directly made to us to improve some of the figures, which we will include in the revision.

---

## Author Response (AR1)

**Reply to reviewers (HESS-2018-81)**

Referee #1

1) *Disclosure and data location*: We have made the changes as outlined in the interactive reply to the reviewer (HESS-2018-81-AC1) and as previously discussed with the Editors. This includes a
5  statement in the Introduction that the exact site location cannot be report due to 'proprietary reasons'. We have included additional details in the general site description, as requested by the Editors.

2) *Dissimilatory nitrate reduction (DNRA)*: We added in the Introduction (p.2, line 33) the equation for DNRA along with those for denitrification and anammox. However, we also concluded (p. 11; line
10  32) that unlike other subtropical estuaries DNRA is unlikely to be a key process in this environment because it cannot account for the simultaneous decline in both $NH_4^+$ and $NO_3^-$ in the profiles. The discrepancy between this and other studies can be accounted for by the unusual 1:1 molar ratio for $NH_4^+$ and $NO_3^-$ in the contamination source, which would suit anammox best.

Minor comments.

15  3.25. *What River?*: Unfortunately, the site owner clearly indicated the name of the river could not be revealed as a condition to release data for publication.

4.5. *Rubber mats*: See our reply in HESS-2018-81-AC1. Essentially, the sampling environment was challenging and we did the best we could do. The interval between sampling points was deliberately kept relatively large to account for potential disturbances during sampling. We have removed the
20  last sentence of this paragraph to avoid confusion.

4.25. *Filtration of nitrate samples:* The water samples for the stable isotopes of nitrate were filtered through a 0.2 μm filter and then frozen.

9.24. *Isotopic signature for groundwater*: A detailed answer was provided in HESS-2018-81-AC1. Essentially, due to widespread contamination at the site it is not clear where 'pristine' groundwater
25  could have been found and building a meteoric water line for the site would require several years of sampling, well beyond our capacity. The use of Sydney rainfall and groundwater as a proxy is reasonable in our opinion as it is close (less than 100 km) from the site and also near the coastline.

Fig. 2. *Show directions to sea*: Done

*Other figures*: Based on the advice from both reviewers, we have updated most of the figures and
30  revised most of the figure captions

Reviewer#2

1) *First study combining intertidal and hyporheic mixing*: We now highlight this at the end of the abstract and in the title

35  2) *Site location*: See response to similar concern by Reviewer #1.

3) *Present oxygen saturation levels in Table 1:* Because of the qualitative nature of this data (that is, it was collected from purge water only), we left as is. The main message is that oxygen levels were low in porewater and a correction for %saturation is not required to demonstrate that.

4) *Salinity vs. chlorinity*: Whilst it is customary to use salinity as a conservative tracer in estuarine environments, it is not appropriate here because the reactive constituents ($NH_4^+$ and $NO_3^-$) concentrations are high enough to influence salinity. Thus, chlorinity was used to evaluate mixing instead.

5) *Figure 1*: We have slightly updated the figure to more strongly emphasise the hyporheic zone.

6) *Figure 3*: The graph was updated as per the reviewer's suggestions.

7) *Figure 4*: Chlorinity now expressed in g/L throughout the manuscript. Figure has been 'decluttered' as much as feasible.

8) *Figures 5 and 6* have been redrawn to remove the confusion

9) *Figure 7*: Comments about the Profile 2 samples included.

10) Figure 8: Caption corrected to 'dashed line'.

11) Figure 9: The meaning of the arrows is now explained

12) Figure 10: The main inference from the figure now explained in the caption.

13) The intrepid reviewer provided three pages of minor revisions...we have agreed with most of these or have clarified the statement in other ways. These changes can be found in track-change in the text. Notable modifications include:

- One requested change not agreed to was to use 'intertidal recirculation' instead of 'seawater recirculation'. The latter term is common usage at present and the former could be confused with the specific process of 'tidal circulation' we referred to elsewhere in the text. However, following Heiss and Michael (2014), tidal recirculation is now 'tide-induced circulation' in the text and on Figure 1.
- Yes, the reviewer read correctly, N concentrations were up to the g/L range. The site is indeed under the close scrutiny of the local Environmental Protection Authority and the (new) owner of the industrial facility is actively trying to address the problem (hence this study).
- No, the reviewer did not get the volume of porewater pumped out correctly. It is 210 mL as stated. We can only assume some confusion with the paragraph below, which is about the processing of the samples, not new volumes collected.
- We have not provided a specific reference for the isotope ratio mass spectrometry of the water samples because it is essentially a routine technique and all the instrumentation used is in the text. The interested readers can easily contact GNS New Zealand (the largest provider for isotopic analyses in the southern hemisphere) for additional details. However, due to the unusual results obtained for $\delta^{18}O$-$H_2O$, we did seek confirmation from the laboratory that the numbers were accurate. They confirmed the results were correct and this gave us the confidence to go ahead with a deeper interpretation.
- We have not included additional information about age-dating of groundwater with radon because we made little use of this information.
- Nitritation is not a typo. It is the production of nitrite.
- Page 11 line 7. I could not see the problem with the Greek symbols on the original document

**Nitrogen attenuation, dilution and recycling in the intertidal hyporheic zone of a subtropical estuary**

**Authors:** Sébastien Lamontagne[1], Frédéric Cosme[2], Andrew Minard[2], and Andrew Holloway[3]

**Affiliations:**

[1]CSIRO Land & Water, PB 2, Glen Osmond 5064, Australia

[2]Golder Associates, Richmond, VIC 3121, Australia

[3]Golder Associates, St Leonards, NSW 2065, Australia

*Correspondence to*: S. Lamontagne (sebastien.lamontagne@csiro.au)

**Abstract.**  Tidal estuarine channels have complex and dynamic interfaces controlled by upland groundwater discharge, waves, tides, and channel velocities that also control biogeochemical processes within adjacent sediments. In an Australian subtropical estuary,  discharging groundwater with elevated (>300 mg N L$^{-1}$) NH$_4^+$ and NO$_3^-$ concentrations indicated  80 % of the N  attenuated at this interface, one of the highest N removal rates (>100 mmol m$^{-2}$ day$^{-1}$) measured for intertidal sediments. The remaining N was also diluted by a factor of two or more by mixing with surface water before being discharged to the estuary. Most of the mixing occurred in a 'hyporheic zone' in the upper 50 cm of the channel bed. However, groundwater entering this zone was already partially mixed (12 – 60 %) with surface water via  tide-induced circulation . Below the hyporheic zone (50 – 125 cm below the channel bed), NO$_3^-$ concentrations declined slightly faster than NH$_4^+$ concentrations and $\delta^{15}N_{NO_3}$ and $\delta^{18}O_{NO_3}$ gradually increased, suggesting a co-occurrence of anammox and denitrification. In the hyporheic zone, $\delta^{15}N_{NO_3}$ continued to become enriched (consistent with either denitrification or anammox) but $\delta^{18}O_{NO_3}$ became more depleted (indicating some nitrification). A high $\delta^{15}N_{NO_3}$ (23 – 35‰) and a low $\delta^{18}O_{NO_3}$ (1.2 – 8.2‰) in all porewater samples indicated that the original synthetic nitrate pool (industrial NH$_4$NO$_3$; $\delta^{15}$N ~ 0‰; $\delta^{18}$O ~ 18 – 20‰) had turned-over  during transport in the aquifer before reaching the channel bed. Whilst porewater NO$_3^-$ was more $\delta^{18}$O depleted than its synthetic source, porewater $\delta^{18}O_{H_2O}$ (–3.2 to –1.8‰) was enriched by 1–4‰ relative to rainfall-derived groundwater mixed with seawater. Isotopic fractionation from H$_2$O uptake during the N cycle and H$_2$O production during synthetic NO$_3^-$ reduction are the probable causes for this $\delta^{18}O_{H_2O}$ enrichment. Whilst occurring at a smaller spatial scale than tide-induced

circulation, hyporheic exchange can provide a similar magnitude of mixing and biogeochemical transformations for groundwater solutes discharging through intertidal zones.

**Keywords:** groundwater – surface water interactions, submarine groundwater discharge, nitrate, isotopic fractionation, hyporheic

**1 Introduction**

In permeable sediments, there is active mixing between surface water and groundwater by hyporheic exchange and seawater recirculation (Jones and Mulholland, 2000;Heiss and Michael, 2014) (Fig. 1). Hyporheic exchange is induced by flows and currents over uneven riverbeds creating zones where surface water moves in and porewater moves out of the sediments (Harvey and Bencala, 1993). In marine environments, tides, wave action and density differences between discharging fresh groundwater and seawater also generate groundwater – surface water mixing (collectively referred to here as 'seawaterseawater recirculation') (Burnett et al., 2003;Sawyer et al., 2013;Precht and Huettel, 2003;Pool et al., 2015). When concentrations are more elevated in groundwater, hyporheic exchange and seawater recirculation can spread a solute load over time and in general will tend to lower concentrations at the discharge point (Li et al., 1999;Murgulet and Tick, 2016). However, because hyporheic exchange and seawater recirculation also bring labile organic matter, oxygen and other compounds to the subsurface (Santos et al., 2011;Ahmerkamp et al., 2017), mixing zones are also very active biogeochemical environments where reactive contaminants like $NH_4^+$ and $NO_3^-$ can be attenuated via a range of biogeochemical processes (Ullman et al., 2003;Abe et al., 2009;Ueda et al., 2003). When attenuation also takes place, both the contaminant concentration and the contaminant load to surface water is reduced by groundwater – surface water exchange.

The mixing and attenuation of $NH_4^+$ and $NO_3^-$ in contaminated groundwater discharging into a subtropical southeast Australian estuary was evaluated by collecting channel riverbed porewater profiles using a drive points (Cranswick et al., 2014). Chloride concentrations were used as a stable tracer to determine the effect of movement and mixingA conservative tracer (chloride) evaluated mixing, $^{222}$Rn estimated residence times (Hoehn and Cirpka, 2006;Lamontagne and Cook, 2007) and various other parameters (including $NH_4^+$ and $NO_3^-$ concentrations and the dual isotopes isotopic composition of $NO_3$ nitrate) evaluated N cycling in the subsurface. The isotopic composition of water was also initially measured to evaluate mixing owing to the large difference in isotopic composition between rainfall-derived groundwater and seawater (Clark and Fritz, 1997). Instead, owing to the large $NO_3^-$ concentrations in this environment, However, the isotopic composition of groundwater was found not to be conservative and was used to further evaluate N attenuation processes. The implications for the evaluation of the N cycle in contaminated aquifers are discussed.For proprietary reasons, details on the exact location of the site are not being reported.

**1.1 Key biogeochemical processes**

In a contaminated aquifer environment, some of the key processes likely to control the N cycle will include nitrification (Casciotti et al., 2010):

$$NH_4^+ + 2O_2 \rightarrow NO_3^- + H_2O + 2H^+ \tag{1},$$

denitrification (here shown via organic matter oxidation; (Schiff and Anderson, 1987)):

$$(CH_2O)_{106}(NH_3)_{16}(H_3PO_4) + 94.4\ NO_3^- + 92.4H^+ \rightarrow 106CO_2 + 55.2\ N_2 + HPO_4^{-2} + 177.2\ H_2O \qquad (2),$$

dissimilatory $NO_3^-$ reduction to $NH_4^+$ (DNRA, here shown via organic matter oxidation; (Schiff and Anderson, 1987)):

$$(CH_2O)_{106}(NH_3)_{16}(H_3PO_4) + 53\ NO_3^- + 120H^+ \rightarrow 106CO_2 + 69\ NH_4^+ + HPO_4^{-2} + 53\ H_2O \qquad (3),$$

and anaerobic ammonium oxidation (anammox; (Brunner et al., 2013)):

$$1.3NO_2^- + NH_4^+ \rightarrow N_2 + 0.3NO_3^- + 2H_2O \qquad (4\cancel{3}).$$

Annamox tends to co-occur with other biogeochemical processes producing $NO_2^-$, such as denitrification (Zhou et al., 2016). Other possible reactions include ion exchange with aquifer materials, the assimilation of $NH_4^+$ and $NO_3^-$ into microbial biomass, , and the mineralisation of organic-N during decomposition (Casciotti, 2016;Appelo and Postma, 1993). All the above biogeochemical reactions are expected to modify the nitrogen ($^{15}$N:$^{14}$N) and oxygen ($^{18}$O:$^{16}$O) isotope ratios in the original $NH_4^+$ and $NO_3^-$ pools via kinetic fractionation and isotopic equilibrium effects (isotopic ratios are generally expressed in parts per thousands (‰) relative to a standard using the del ($\delta$) notation or, for $\delta^{15}$N, ($^{15}$N:$^{14}$N$_{sample}$/$^{15}$N:$^{14}$N$_{standard}$ − 1) · 1000). For example, a $NO_3^-$ pool undergoing denitrification will become more enriched in its heavier isotopes as the lighter ones are selectively removed. The enrichment factor for $\delta^{15}N_{NO_3}$ during denitrification ($^{15}\varepsilon_{NO_3 \to N_2}$) has been found to vary from $9 - 20$‰ and the one for $\delta^{18}O_{NO_3}$ ($^{18}\varepsilon_{NO_3 \to N_2}$) from $4 - 16$‰ (Knoller et al., 2011;Bottcher et al., 1990;Dahnke and Thamdrup, 2013;Wenk et al., 2014). Anammox also strongly fractionates $^{15}$N in the $NH_4^+$, $NO_2^-$ and $NO_3^-$ pools present via kinetic and isotopic equilibrium effects (Brunner et al., 2013). However, the systematics for oxygen fractionation during anammox are not known (Casciotti, 2016). Nitrification is a special case because the $\delta^{18}$O signature of the $NO_3^-$ produced will be a function of the isotopic signature of the ambient $O_2$ and $H_2O$ (Mayer et al., 2001;Snider et al., 2010;Casciotti et al., 2010). Synthetically produced $NO_3^-$ tends to be $^{18}$O-enriched relative to $NO_3^-$ produced via nitrification because  all the oxygen is atmospheric in origin ($\delta^{18}O_{O_2} \sim 23$‰) whereas during nitrification two out of three O originates from water, which is generally $^{18}$O-depleted ($\delta^{18}O_{H_2O} < 5$‰) relative to atmospheric $O_2$ (Mengis et al., 2001).

**1.2 Terminology**

The following terms are used to define either different sources of water or exchange processes in the profiles. *Porewater* is used for any water recovered in the subsurface, regardless of its origin. *Terrestrial groundwater* is used for groundwater originating from rainfall recharge before any significant mixing with estuarine water has occurred. The *hyporheic zone* is defined as the upper part of the channel bed where surface and subsurface water mix because of processes such as currents, wave pumping or any other. *Tide-induced circulation* is the process by which estuarine water tends to move inland over the freshwater table during the rising tide and discharges back to the estuary during the falling tide. *Surface water* represents the estuary. When describing the profiles, porewater from below the hyporheic zone is further referred to as *groundwater* while porewater within the hyporheic zone is further referred to as *hyporheic water*.

**2 Methods**

**2.1 Site description**

The site is located in the estuarine section of a large river on the southeast coast of Australia. The flow regime is similar to other large rivers in this region, with occasional floods flushing the estuary with freshwater and prolonged low-flow periods resulting in seawater-like salinities near the mouth. The tidal amplitude in the lower
5   estuary is similar to the ocean (1 – 2 m). The climate is subtropical with precipitation (~800 mm) lower than potential evaporation (~1730 mm). Land-use in the catchment is also typical for southeast Australia, including conservation areas, farming and mining in the headwaters and a mixture of urban, industrial and conservation areas in the estuary, . The site itself is located near an industrial facility on partially reclaimed land. Groundwater $NH_4^+$ and $NO_3^-$ concentrations are elevated in and near the industrial
10   facility (>5000 mg N $L^{-1}$ at some locations). The groundwater contamination is widespread  and may have several sources. In other words, there is not a single contamination point and associated groundwater plume downgradient. However, the most impacted area is located on the south-eastern side of the site and the associated discharge point along the estuary is known. This area has been instrumented with nested piezometers transects in the four hydrostratigraphic units present including the uppermost units 1 and 2, the two most likely to outcrop in
15   the intertidal zone.

      Three drive point profiles were collected in the intertidal zone in the vicinity of the main impacted area (Fig. 2). Profile 2 was located in the alignment of the transect of nested piezometers described above, whilst profiles 1 and 3 were located approximately  100 m south and north from Profile 2, respectively. The intertidal zone at the site consists of a steep artificial rock embankment abutting a silty sand channel bed
20   interspersed with oyster beds on harder substrates. The channel bed would typically  be exposed only for a few hours at each low tide. Sampling occurred on the afternoon of 27 April 2017 and was planned to coincide with the minimum monthly low tide level . Profile 1 was collected at the end of the ebbing tide, Profile 2 at low tide, and Profile 3 during the beginning of the flood tide. The sampling locations were 2 – 5 m offshore  the rock embankment (to prevent interference from
25   buried rocks) and in approximately 1 – 10 cm of surface water. Rubber mats were deployed on the channel bed around the drive points to minimise disturbance during sampling.

      The profiles were collected using a drive point system designed to collect sediment porewater at up to 1.25 m  below ground surface. The drive point consisted of a 1.5 m x 24 mm outer
30   diameter stainless steel tube to which  a 10 cm drive point head was attached. The drive point head had a 5 cm screen and was 
[revised manuscript text omitted]
 the values associated with average annual volume-weighed precipitation ($\delta^2H$ = –20.2‰ and $\delta^{18}O$ = –4.50‰) and average annual  volume-weighed winter precipitation ($\delta^2H$ = –33.0‰ and

5    $\delta^{18}O$ = –6.24‰) for Sydney ( Hughes and Crawford, 2013), or.  the isotopic signature for shallow groundwater in Sydney ($\delta^2H$ = –22.9‰ and $\delta^{18}O$ = –4.77‰; (Hughes and Crawford, 2013)). The comparison of chloride and $\delta^2H_{H_2O}$ shows that the porewater samples were within expectations for mixing between two water sources (estuarine water and terrestrial groundwater ), especially if the groundwater end-

10   member was more similar to winter Sydney rainfall (Fig. 7a). However, when looking at chloride and $\delta^{18}O_{H_2O}$ (Fig. 7b), porewater samples from profiles 2 and 3 were at least 1 – 4‰ enriched relative to conservative mixing lines and more similar to annual than winter Sydney rainfall. The discrepancy was noticeable for profile 2 samples, especially when expressed on a $\delta^2H$-$\delta^{18}O$ plot (Fig. 8). Water table evaporation can shift the isotopic composition of groundwater to the right of the meteoric water line (Clark and Fritz, 1997). However, evaporation would enrich

15   both $\delta^2H_{H_2O}$ and $\delta^{18}O_{H_2O}$ whereas (relative to Sydney groundwater), Profile 2 appeared $\delta^{18}O_{H_2O}$ enriched and possibly slightly $\delta^2H_{H_2O}$ depleted. As Profile 2 is aligned with what is thought to be one of the most impacted groundwater flow lines for the site, the apparent shift in the isotopic composition of water may be related to nitrogen cycling during transport with the aquifer.

        There is also some evidence for non-conservative mixing in the isotopic composition of water at the scale

20   of the profiles. In Profile 2, there was a gradual 1.4‰ shift in $\delta^{18}O_{H_2O}$ upwards once mixing was 
[revised manuscript text omitted]

These findings appear at odds with other studies from tropical and subtropical estuaries suggesting DNRA is a dominant $NO_3^-$ removal processes in sediments (Dunn et al., 2012;Dunn et al., 2013;Dong et al., 2011). Some level of DNRA is likely in this environment and would similarly contribute to the observed trends in the $NO_3^-$ isotopes,

15  isotopes. However, DNRA cannot account for the similar variations in $NH_4^+$ and $NO_3^-$ concentrations in the profiles. If DNRA was dominant, $NO_3^-$ would decline whilst $NH_4^+$ would increase. This discrepancy with other subtropical estuaries can be attributed to the nature of the groundwater N contamination source, where a large 1:1 molar input of $NH_4^+$ and $NO_3^-$ would favour anammox over DNRA.

20  **4.2 Stable isotopes of water**

[revised manuscript text omitted]

**Figure 4.** Intertidal zone pPorewater profiles for selected parameters collected in the intertidal zone. Based on the trends in chloride (A) and the stable isotopes of water (B – C), two scales of groundwater – surface water mixing were apparent in the profiles. In the top 50 cm of the profiles, a gradual change in concentration between a surface water and deeper porewater end-member is consistent with hyporheic mixing. Below the hyporheic zone, chloride concentrations were relatively constant but were intermediate between terrestrial groundwater and surface water, suggesting return flow from tide-induced circulation. See text for explanations about the other parameters.

[Figure]

**Figure 5.** Evaluation of mixing and transformations for $NH_4^+$ ( ) and $NO_3^-$ ( ) for Profile 1 (top ), Profile 2 (middle ) and Profile 3 (bottom ). The vertical pink lines represent samples collected in the 'groundwater' zone and those to the right of this line are within the hyporheic zone, based on chloride concentrations. The blue circles represent the surface water samples.

[Figure]

**Figure 6.** Evaluation of mixing and transformations for $\delta^{15}N_{NO_3}$ (A, C, E) and $\delta^{18}O_{NO_3}$ (B, D, F) for Profile 1 (top row), Profile 2 (middle row) and Profile 3 (bottom row). The vertical pink lines represent samples collected in the 'groundwater' zone and those to the right of this line are within the hyporheic zone, based on chloride concentrations. The blue circles represent the surface water samples. The black solid lines represent expected isotopic values if only conservative mixing occurred in the hyporheic zone. The red arrows indicate the position of samples in the profiles (from bottom to top), highlighting an apparent reversal in the path of isotopic enrichment for $NO_3^-$ at the base of the hyporheic zone.

[Figure]

**Figure 7.** Chloride-based mixing lines for the isotopic composition of porewater relative to surface water and  annual Sydney rainfall orand winter Sydney rainfall. Sydney rainfall and groundwater wereas used as  proxiesy for the isotopic composition of unimpacted groundwater at the site.  Profile 2 porewater $\delta^{18}O_{H_2O}$ values cannot be readily accounted for by a two end-member conservative mixing model for terrestrial groundwater and surface water.

[Figure]

**Figure 8.** Isotopic composition for surface water, porewater, and Sydney groundwater and rainfall relative to the meteoric water line for Sydney (Lucas Height; solid line). Dashed lines represent potential mixing lines between terrestrial groundwater and surface water. Note that evaporation lines for groundwater would be very similar to the mixing line in this environment. Profile 2 porewater samples do not conform to a two-end member conservative mixing model between terrestrial groundwater and surface water.

[Figure]

**Figure 9.** Mixing model for the isotopic composition of water for Profile 2, showing a small depletion trend upward in the profiles for $\delta^{18}O_{H_2O}$ (A) but not $\delta^2H_{H_2O}$ (B). The red arrows indicate the position of the samples in the profile, from bottom to top, highlighting the depletion trend for $\delta^{18}O_{H_2O}$ is continuous. When corrected for mixing, $\delta^{18}O_{H_2O}$ would be –3.2‰ at the top of the profile, representing a –1.4‰ shift relative to the base of the profile.

[Figure]

**Figure 10.** Variations in $\delta^{15}N_{NO_3}$ and $\delta^{18}O_{NO_3}$ in intertidal porewater, surface water, one synthetic NH₄NO₃ sample from the plant and three groundwater samples collected underneath the plant (A. Minard, *unpublished data*). The enrichment in $\delta^{15}N_{NO_3}$ and depletion in $\delta^{18}O_{NO_3}$ in porewater relative to the industrial source suggest a complete turnover of the nitrate pool during groundwater transport.